# TOWARDS UNDERSTANDING THE SPECTRAL BIAS OF DEEP LEARNING

## ABSTRACT

An intriguing phenomenon observed during training neural networks is the spectral bias, where neural networks are biased towards learning less complex functions. The priority of learning functions with low complexity might be at the core of explaining generalization ability of neural network, and certain efforts have been made to provide theoretical explanation for spectral bias. However, there is still no satisfying theoretical result justifying the underlying mechanism of spectral bias. In this paper, we give a comprehensive and rigorous explanation for spectral bias and relate it with the neural tangent kernel function proposed in recent work. We prove that the training process of neural networks can be decomposed along different directions defined by the eigenfunctions of the neural tangent kernel, where each direction has its own convergence rate and the rate is determined by the corresponding eigenvalue. We then provide a case study when the input data is uniformly distributed over the unit sphere, and show that lower degree spherical harmonics are easier to be learned by over-parameterized neural networks.

## 1 INTRODUCTION

Over-parameterized neural networks have achieved great success in many applications such as computer vision (He et al., 2016), natural language processing (Collobert and Weston, 2008) and speech recognition (Hinton et al., 2012). It has been shown that over-parameterized neural networks can fit complicated target function or even randomly labeled data (Zhang et al., 2016) and still exhibit good generalization performance when trained with real labels. Intuitively, this is at odds with the traditional notion of generalization ability such as model complexity. In order to understand neural network training, a line of work (Soudry et al., 2017; Gunasekar et al., 2018b;a) has made efforts in the perspective of "implicit bias", which states that training algorithms for deep learning implicitly pose an inductive bias onto the training process and lead to a solution with low complexity measured by certain norms in the parameter space of the neural network.

Among many attempts to establish implicit bias, Rahaman et al. (2018) pointed out an intriguing phenomenon called *spectral bias*, which says that during training, neural networks tend to learn the components of lower complexity faster. The concept of spectral bias is appealing because this may intuitively explain why over-parameterized neural networks can achieve a good generalization performance without overfitting. During training, the networks fit the low complexity components first and thus lie in the concept class of low complexity. Arguments like this may lead to rigorous guarantee for generalization.

Great efforts have been made in seek of explanations about the spectral bias. Rahaman et al. (2018) evaluated the Fourier spectrum of ReLU networks and empirically showed that the lower frequencies are learned first; also lower frequencies are more robust to random perturbation. Andoni et al. (2014) showed that for a sufficiently wide two-layer network, gradient descent with respect to the second layer can learn any low degree bounded polynomial. Xu (2018) provided Fourier analysis to two-layer networks and showed similar empirical results on one-dimensional functions and real data. Nakkiran et al. (2019) used information theoretical approach to show that networks obtained by stochastic gradient descent can be explained by a linear classifier during early training. All these studies provide certain explanations about why neural networks exhibit spectral bias in real tasks. But explanations in the theoretical aspect, if any, are to some extent restricted. For example, the popular Fourier analysis is usually done in the one-dimensional setting, and thus lacks generality.

Meanwhile, a recent line of work (Jacot et al., 2018; Du et al., 2018b; Li and Liang, 2018; Chizat and Bach, 2018) has shed light on new approaches to analyze neural networks. In particular, they show that under certain over-parameterized condition, the neural network trained by gradient descent behaves similar to the kernel regression predictor using the neural tangent kernel (NTK) (Jacot et al., 2018). Du et al. (2018b) showed that the convergence is provably guaranteed under certain over-parameterization conditions determined by the smallest eigenvalue of NTK. Arora et al. (2019b) further gave a finer characterization of error convergence based on the eigenvalues of NTK's Gram matrix. Su and Yang (2019) improved the convergence guarantee in terms of the $k$-th largest eigenvalue for certain target functions.

Inspired by these works mentioned above, we can present a theoretical explanation for spectral bias. Under NTK regime, we establish a precise characterization for the training process of neural networks. More specifically, we theoretically prove that over-parameterized neural networks' training process can be controlled by the eigenvalues of the integrating operator defined by the NTK. Under the specific case of uniform distribution on unit sphere, we give an exact calculation for these eigenvalues and show that the lower frequencies have larger eigenvalues, which thus leads to faster convergence. We also conduct experiments to corroborate the theory we establish.

Our contributions are highlighted as follows:

1. We prove a generic theorem for arbitrary data distributions, which states that under certain sample complexity and over-parameterization conditions, the error term's convergence along different directions actually relies on the corresponding eigenvalues. This theorem gives a more precise control on the regression residual than Su and Yang (2019), where the authors focused on the case when the labeling function is close to the subspace spanned by the first few eigenfunctions.

2. We present a more general result about the spectra of the neural tangent kernel. In particular, we show that the order of eigenvalues appears as $\mu_k = \Omega(\max\{k^{-d-1}, d^{-k+1}\})$. Our result is better than the bound $\Omega(k^{-d-1})$ derived in Bietti and Mairal (2019) when $d \gg k$, which is clearly a more practical setting.

3. We establish a rigorous explanation for the spectral bias based on the aforementioned theoretical results without any specific assumptions on the target function. We show that the error terms from different frequencies are provably controlled by the eigenvalues of the NTK, and the lower-frequency components can be learned with less training examples and narrower networks with a faster convergence rate. As far as we know, this is the first attempt to give a comprehensive theory justifying the existence of spectral bias.

## 1.1 ADDITIONAL RELATED WORK

Recently, there is a rich literature about the property of neural tangent kernel. Jacot et al. (2018) first showed that during training, the network function follows a descent along the kernel gradient with respect to the Neural Tangent Kernel (NTK) under infinity width setting. Li and Liang (2018) and Du et al. (2018b) implicitly built connection between Neural Tangent Kernel and gradient descent by showing that GD can provably optimize sufficiently wide two-layer neural networks. In Du et al. (2018b), it is shown that gradient descent can achieve zero training loss at a linear convergence rate for training two-layer ReLU network with square loss. Allen-Zhu et al. (2018); Du et al. (2018a); Zou et al. (2018); Cao and Gu (2019b); Arora et al. (2019a); Zou and Gu (2019); Cao and Gu (2019a) further studied the optimization and generalization of deep neural networks. These papers are all in the so-called neural tangent kernel regime, and their requirements on the network width depend either implicitly or explicitly on the smallest eigenvalue of the kernel Gram matrix. Later, Su and Yang (2019) showed that this smallest eigenvalue actually scales in the number of samples $n$ and will eventually converge to 0. In order to obtain constant convergence rate, Su and Yang (2019) assumed that the target function $f^*$ can be approximated by the first few eigenfunctions of the integrating operator $L_\kappa f(s) := \int_{\mathbb{S}^d} \kappa(x, s) f(s) d\tau(s)$ where $\kappa(\cdot, \cdot)$ is the NTK function and $\tau(s)$ is the input distribution, and proved linear convergence rate up to the this approximation error.

A few theoretical results have been established towards understanding the spectra of neural tangent kernels. Bach (2017) studied two-layer ReLU networks by relating it to kernel methods, and proposed a harmonic decomposition for the functions in the reproducing kernel Hilbert space which we utilize in our proof. Based on the technique in Bach (2017), Bietti and Mairal (2019) studied the eigenvalue decay of integrating operator $L_\kappa f(x)$ defined by NTK on unit sphere by using spheri-

cal harmonics. Vempala and Wilmes (2018) calculated the eigenvalues of NTK corresponding to two-layer neural networks with sigmoid activation function. Basri et al. (2019) established similar results as Bietti and Mairal (2019), but considered the case of training the first layer parameters of a two-layer networks with bias terms. Yang and Salman (2019) studied the the eigenvalues of integral operator with respect to the NTK on Boolean cube by Fourier analysis.

The rest of the paper is organized as follows. We state the notation, problem setup and other preliminaries in Section 2 and present our main results in Section 3. In Section 4, we present experimental results to support our theory. Proofs of our main results can be found in the appendix.

## 2 PRELIMINARIES

In this section we introduce the basic problem setup including the neural network structure and the training algorithm, as well as some background on the neural tangent kernel proposed recently in Jacot et al. (2018) and the corresponding integral operator.

### 2.1 NOTATION

We use lower case, lower case bold face, and upper case bold face letters to denote scalars, vectors and matrices respectively. For a vector $\mathbf{v} = (v_1, \ldots, v_d)^T \in \mathbb{R}^d$ and a number $1 \leqslant p < \infty$, we denote its $p-$norm by $\|\mathbf{v}\|_p = (\sum_{i=1}^d |v_i|^p)^{1/p}$. We also define infinity norm by $\|\mathbf{v}\|_\infty = \max_i |v_i|$. For a matrix $\mathbf{A} = (A_{i,j})_{m \times n}$, we use $\|\mathbf{A}\|_0$ to denote the number of non-zero entries of $\mathbf{A}$, and use $\|\mathbf{A}\|_F = (\sum_{i,j=1}^d A_{i,j}^2)^{1/2}$ to denote its Frobenius norm. Let $\|\mathbf{A}\|_p = \max_{\|\mathbf{v}\|_p \leqslant 1} \|\mathbf{A}\mathbf{v}\|_p$ for $p \geqslant 1$, and $\|\mathbf{A}\|_{\max} = \max_{i,j} |A_{i,j}|$. For two matrices $\mathbf{A}, \mathbf{B} \in \mathbb{R}^{m \times n}$, we define $\langle \mathbf{A}, \mathbf{B} \rangle = \mathrm{Tr}(\mathbf{A}^\top \mathbf{B})$. We use $\mathbf{A} \succeq \mathbf{B}$ if $\mathbf{A} - \mathbf{B}$ is positive semi-definite. In addition, we define the asymptotic notations $\mathcal{O}(\cdot)$, $\widetilde{\mathcal{O}}(\cdot)$, $\Omega(\cdot)$ and $\widetilde{\Omega}(\cdot)$ as follows. Suppose that $a_n$ and $b_n$ be two sequences. We write $a_n = \mathcal{O}(b_n)$ if $\limsup_{n\to\infty} |a_n/b_n| < \infty$, and $a_n = \Omega(b_n)$ if $\liminf_{n\to\infty} |a_n/b_n| > 0$. We use $\widetilde{\mathcal{O}}(\cdot)$ and $\widetilde{\Omega}(\cdot)$ to hide the logarithmic factors in $\mathcal{O}(\cdot)$ and $\Omega(\cdot)$.

### 2.2 PROBLEM SETUP

Here we introduce the basic problem setup. We consider two-layer fully connected neural networks of the form

$$f_{\mathbf{W}}(\mathbf{x}) = \sqrt{m} \cdot \mathbf{W}_2 \sigma(\mathbf{W}_1 \mathbf{x}),$$

where $\mathbf{W}_1 \in \mathbb{R}^{m \times (d+1)}$, $\mathbf{W}_2 \in \mathbb{R}^{1 \times m}$[1] are the first and second layer weight matrices respectively, and $\sigma(\cdot) = \max\{0, \cdot\}$ is the entry-wise ReLU activation function. The network is trained according to the square loss on $n$ training examples $S = \{(\mathbf{x}_i, y_i) : i \in [n]\}$:

$$L_S(\mathbf{W}) = \frac{1}{n} \sum_{(\mathbf{x}_i, y_i) \in S} (y_i - \theta f_{\mathbf{W}}(\mathbf{x}_i))^2,$$

where $\theta$ is a small coefficient to control the effect of initialization, and the data inputs $\{\mathbf{x}_i\}_{i=1}^n$ is assumed to follow some unknown distribution $\tau$ on the unit sphere $\mathbb{S}^d \in \mathbb{R}^{d+1}$. Without loss of generality, we also assume that $|y_i| \leqslant 1$.

We first randomly initialize the parameters of the network, and then apply gradient descent to optimize both layers. We present our detailed neural network training algorithm in Algorithm 1.

---

**Algorithm 1** GD for DNNs starting at Gaussian initialization

---

**Input:** Number of iterations $T$, step size $\eta$.
Generate each entry of $\mathbf{W}_1^{(0)}$ and $\mathbf{W}_2^{(0)}$ from $N(0, 2/m)$ and $N(0, 1/m)$ respectively.
**for** $t = 0, 1, \ldots, T-1$ **do**
    Update $\mathbf{W}^{(t+1)} = \mathbf{W}^{(t)} - \eta \cdot \nabla_{\mathbf{W}} L_S(\mathbf{W}^{(t)})$.
**end for**
**Output:** $\mathbf{W}^{(T)}$.

---

[1] Here the dimension of input is $d + 1$ since throughout this paper we assume that all training data lie in the $d$-dimensional unit sphere $\mathbb{S}^d \in \mathbb{R}^{d+1}$.

The initialization scheme for $\mathbf{W}^{(0)}$ given in Algorithm 1 is known as He initialization (He et al., 2015). This scheme generates each entry of the weight matrices from a Gaussian distribution with mean zero. The variances of the Gaussian distributions in initialization are chosen following the principle that the initialization does not change the magnitudes of inputs in each layer. The second layer parameter is not associated with the ReLU activation function, thus it is initialized with variance $1/m$ instead of $2/m$.

### 2.3 Neural Tangent Kernel

Many attempts have been made to study the convergence of gradient descent assuming the width of the network is extremely large (Du et al., 2018b; Li and Liang, 2018). When the width of the network goes to infinity, with certain initialization on parameters, the inner product of gradients of the output function would converge to a limiting kernel, i.e., Neural Tangent Kernel (Jacot et al., 2018). In this paper, we denote it by $\kappa(\mathbf{x}, \mathbf{x}') = \lim_{m \to \infty} m^{-1} \langle \nabla_{\mathbf{W}} f_{\mathbf{W}^{(0)}}(\mathbf{x}), \nabla_{\mathbf{W}} f_{\mathbf{W}^{(0)}}(\mathbf{x}') \rangle$ and we have

$$\kappa(\mathbf{x}, \mathbf{x}') = \langle \mathbf{x}, \mathbf{x}' \rangle \cdot \kappa_1(\mathbf{x}, \mathbf{x}') + 2 \cdot \kappa_2(\mathbf{x}, \mathbf{x}'), \tag{2.1}$$

where

$$\begin{aligned}
\kappa_1(\mathbf{x}, \mathbf{x}') &= \mathbb{E}_{\mathbf{w} \sim N(\mathbf{0}, \mathbf{I})}[\sigma'(\langle \mathbf{w}, \mathbf{x} \rangle)\sigma'(\langle \mathbf{w}, \mathbf{x}' \rangle)], \\
\kappa_2(\mathbf{x}, \mathbf{x}') &= \mathbb{E}_{\mathbf{w} \sim N(\mathbf{0}, \mathbf{I})}[\sigma(\langle \mathbf{w}, \mathbf{x} \rangle)\sigma(\langle \mathbf{w}, \mathbf{x}' \rangle)].
\end{aligned} \tag{2.2}$$

Since we apply gradient descent to both layers, the Neural Tangent Kernel is the sum of two different kernel functions and clearly it can be reduced to one layer training setting. These two kernels are arc-cosine kernels of degree 0 and 1 (Cho and Saul, 2009). Their explicit expressions are given as

$$\kappa_1(t) = \frac{1}{2\pi}\left(\pi - \arccos(t)\right), \quad \kappa_2(t) = \frac{1}{2\pi}\left(t \cdot (\pi - \arccos(t)) + \sqrt{1 - t^2}\right), \tag{2.3}$$

where $t = \langle \mathbf{x}, \mathbf{x}' \rangle / (\|\mathbf{x}\| \|\mathbf{x}'\|)$.

### 2.4 Integral Operator

The theory of integral operator with respect to kernel function has been well studied in machine learning (Smale and Zhou, 2007; Rosasco et al., 2010) thus we only give a brief introduction here. Let $L_\tau^2(X)$ be the Hilbert space of square-integrable functions with respect to a Borel measure $\tau$ from $X \to \mathbb{R}$. For any continuous kernel function $\kappa : X \times X \to \mathbb{R}$ and $\tau$ we can define an integral operator $L_\kappa$ on $L_\tau^2(X)$ by

$$L_\kappa(f)(\mathbf{x}) = \int_X \kappa(\mathbf{x}, \mathbf{y}) f(\mathbf{y}) d\tau(\mathbf{y}), \quad \mathbf{x} \in X. \tag{2.4}$$

It has been pointed out in Cho and Saul (2009) that arc-cosine kernels are positive semi-definite. Thus the kernel function $\kappa$ defined by (2.1) is positive semi-definite being a product and a sum of positive semi-definite kernels. Clearly this kernel is also continuous and symmetric. Thus we know that the neural tangent kernel $\kappa$ is a Mercer kernel.

## 3 Main Results

In this section we present our main results. In Section 3.1, we give a general result on the convergence rate of gradient descent along different eigendirections of neural tangent kernel. Motivated by this result, in Section 3.2, we give a case study on the spectrum of $L_\kappa$ when the input data are uniformly distributed over the unit sphere $\mathbb{S}^d$. In Section 3.3, we combine the spectrum analysis with the general convergence result to give explicit convergence rate for uniformly distributed data on the unit sphere.

### 3.1 Convergence Analysis of Gradient Descent

In this section we study the convergence of Algorithm 1. Instead of studying the standard convergence of loss function value, we aim to provide a refined analysis on the speed of convergence along different directions defined by the eigenfunctions of $L_\kappa$. We first introduce the following definitions and notations.

Let $\{\lambda_i\}_{i \geqslant 1}$ with $\lambda_1 \geqslant \lambda_2 \geqslant \cdots$ be the strictly positive eigenvalues of $L_\kappa$, and $\phi_1(\cdot), \phi_2(\cdot), \ldots$ be the corresponding orthonormal eigenfunctions. Set $\mathbf{v}_i = n^{-1/2}(\phi_i(\mathbf{x}_1), \ldots, \phi_i(\mathbf{x}_n))^\top, i = 1, 2, \ldots$. Note that $L_\kappa$ may have eigenvalues with multiplicities larger than 1 and $\lambda_i, i \geqslant 1$ are not distinct. Therefore for any integer $k$, we define $r_k$ as the sum of the multiplicities of the first $k$ distinct eigenvalues of $L_\kappa$. Define $\mathbf{V}_{r_k} = (\mathbf{v}_1, \ldots, \mathbf{v}_{r_k})$. By definition, $\mathbf{v}_i, i \in [r_k]$ are rescaled restrictions of orthonormal functions in $L_\tau^2(\mathbb{S}^d)$ on the training examples. Therefore we can expect them to form a set of almost orthonomal bases in the vector space $\mathbb{R}^n$. The following lemma follows by standard concentration inequality.

**Lemma 3.1.** Suppose that $|\phi_i(\mathbf{x})| \leqslant M$ for all $\mathbf{x} \in \mathbb{S}^d$ and $i \in [r_k]$. For any $\delta > 0$, with probability at least $1 - \delta$,

$$\|\mathbf{V}_{r_k}^\top \mathbf{V}_{r_k} - \mathbf{I}\|_{\max} \leqslant CM^2\sqrt{\log(r_k/\delta)/n},$$

where $C$ is an absolute constant.

Denote $\mathbf{y} = (y_1, \ldots, y_n)^\top$ and $\widehat{\mathbf{y}}^{(t)} = \theta \cdot (f_{\mathbf{W}^{(t)}}(\mathbf{x}_1), \ldots, f_{\mathbf{W}^{(t)}}(\mathbf{x}_n)), t = 0, \ldots, T$. Then Lemma 3.1 shows that the convergence rate of $\|\mathbf{V}_{r_k}^\top(\mathbf{y} - \widehat{\mathbf{y}}^{(t)})\|_2$ roughly represents the speed gradient descent learns the components of the target function corresponding to the first $r_k$ eigenvalues. The following theorem gives the convergence guarantee of $\|\mathbf{V}_{r_k}^\top(\mathbf{y} - \widehat{\mathbf{y}}^{(t)})\|_2$.

**Theorem 3.2.** Suppose $|\phi_j(\mathbf{x})| \leqslant M$ for $j \in [r_k]$ and $\mathbf{x} \in \mathbb{S}^d$. For any $\epsilon, \delta > 0$ and integer $k$, if $n \geqslant \widetilde{\Omega}(\epsilon^{-2} \cdot \max\{(\lambda_{r_k} - \lambda_{r_k+1})^{-2}, M^4 r_k^2\}), m \geqslant \widetilde{\Omega}(\mathrm{poly}(T, \lambda_{r_k}^{-1}, \epsilon^{-1}))$, then with probability at least $1 - \delta$, Algorithm 1 with $\eta = \widetilde{\mathcal{O}}((m\theta^2)^{-1}), \theta = \widetilde{\mathcal{O}}(\epsilon)$ satisfies

$$n^{-1/2} \cdot \|\mathbf{V}_{r_k}^\top(\mathbf{y} - \widehat{\mathbf{y}}^{(T)})\|_2 \leqslant 2(1 - \lambda_{r_k})^T \cdot n^{-1/2} \cdot \|\mathbf{V}_{r_k}^\top \mathbf{y}\|_2 + \epsilon.$$

**Remark 3.3.** Theorem 3.2 theoretically reveals the spectral bias of deep learning. Specifically, as long as the network is wide enough and the sample size is large enough, gradient descent first learns the target function along the the eigendirections of neural tangent kernel with larger eigenvalues, and then learns the rest components corresponding to smaller eigenvalues. Moreover, by showing that learning the components corresponding to larger eigenvalues can be done with smaller sample size and narrower networks, our theory pushes the study of neural networks in the NTK regime towards a more practical setting. For these reasons, we believe that Theorem 3.2 to certain extent explains the empirical observations given in Rahaman et al. (2018), and demonstrates that the difficulty of a function to be learned by neural network can be characterized in the eigenspace of neural tangent kernel: if the target function has a component corresponding to a small eigenvalue of neural tangent kernel, then learning this function up to good accuracy takes longer time, and requires more examples and a wider network.

### 3.2 SPECTRAL ANALYSIS OF NEURAL TANGENT KERNEL FOR UNIFORM DISTRIBUTION

After presenting a general theorem (without assumptions on data distribution) in the previous subsection, we now study the case when the data inputs are uniformly distributed over the unit sphere. We present our results (an extension of **Proposition 5** in Bietti and Mairal (2019)) of spectral analysis of neural tangent kernel. We show Mercer decomposition of neural tangent kernel for two-layer setting. We give explicit expression of eigenvalues and show orders of eigenvalues in both cases when $d \gg k$ and $k \gg d$.

**Theorem 3.4.** For any $\mathbf{x}, \mathbf{x}' \in \mathbb{S}^d \subset \mathbb{R}^{d+1}$, we have the Mercer decomposition of the neural tangent kernel $\kappa : \mathbb{S}^d \times \mathbb{S}^d \to \mathbb{R}$,

$$\kappa(\mathbf{x}, \mathbf{x}') = \sum_{k=0}^{\infty} \mu_k \sum_{j=1}^{N(d,k)} Y_{k,j}(\mathbf{x}) Y_{k,j}(\mathbf{x}'), \tag{3.1}$$

where $Y_{k,j}$ for $j = 1, \cdots, N(d, k)$ are linearly independent spherical harmonics of degree $k$ in $d+1$ variables with $N(d, k) = \frac{2k+d-1}{k}\binom{k+d-2}{d-1}$ and orders of $\mu_k$ are given by

$\mu_0 = \mu_1 = \Omega(1), \mu_k = 0, k = 2j + 1,$

$\mu_k = \Omega\left(\max\left\{d^{d+1}k^{k-1}(k+d)^{-k-d}, d^{d+1}k^k(k+d)^{-k-d-1}, d^{d+2}k^{k-2}(k+d)^{-k-d-1}\right\}\right), k = 2j,$

where $j \in \mathbb{N}^+$. More specifically, we have $\mu_k = \Omega\left(k^{-d-1}\right)$ when $k \gg d$ and $\mu_k = \Omega\left(d^{-k+1}\right)$ when $d \gg k, k = 2, 4, 6, \ldots$.

**Remark 3.5.** In the above theorem, the coefficients $\mu_k$ are actually different eigenvalues of the integral operator $L_\kappa$ on $L^2_{\tau_d}(\mathbb{S}^d)$ defined by

$$L_\kappa(f)(\mathbf{y}) = \int_{\mathbb{S}^d} \kappa(\mathbf{x}, \mathbf{y}) f(\mathbf{x}) d\tau_d(\mathbf{x}), \quad f \in L^2_{\tau_d}(\mathbb{S}^d),$$

where $\tau_d$ is the uniform probability measure on unit sphere $\mathbb{S}^d$. Therefore the $\lambda_{r_k}$ in Theorem 3.2 is just $\mu_{k-1}$ given in Theorem 3.4 when $\tau_d$ is uniform distribution.

**Remark 3.6.** Vempala and Wilmes (2018) studied two-layer neural networks with sigmoid activation function, and established guarantees to achieve $\epsilon_0 + \epsilon$ error with iteration complexity $T = (d+1)^{\mathcal{O}(k) \log(\|f^*\|_2/\epsilon)}$ under the over-parameterization condition $m = (d+1)^{\mathcal{O}(k)\mathrm{poly}(\|f^*\|_2/\epsilon)}$, where $f^*$ is the target function, and $\epsilon_0$ is certain function approximation error. Another highly related work is Bietti and Mairal (2019), which gives $\mu_k = \Omega(k^{-d-1})$. The order of eigenvalues we present appears as $\mu_k = \Omega(\max(k^{-d-1}, d^{-k+1}))$. This is better when $d \gg k$, which is closer to the practical setting.

### 3.3 EXPLICIT CONVERGENCE RATE FOR UNIFORMLY DISTRIBUTED DATA

In this subsection, we combine our results in the previous two subsections and give explicit convergence rate for uniformly distributed data on the unit sphere. Note that the first $k$ distinct eigenvalues of NTK have spherical harmonics up to degree $k - 1$ as eigenfunctions.

**Corollary 3.7.** Suppose that $k \gg d$, and the sample $\{\mathbf{x}_i\}_{i=1}^n$ follows the uniform distribution $\tau_d$ on the unit sphere $\mathbb{S}^d$. For any $\epsilon, \delta > 0$ and integer $k$, if $n \geqslant \widetilde{\Omega}(\epsilon^{-2} \cdot \max\{k^{2d+2}, k^{2d-2}r_k^2\})$, $m \geqslant \widetilde{\Omega}(\mathrm{poly}(T, k^{d+1}, \epsilon^{-1}))$, then with probability at least $1 - \delta$, Algorithm 1 with $\eta = \widetilde{\mathcal{O}}((m\theta^2)^{-1})$, $\theta = \widetilde{\mathcal{O}}(\epsilon)$ satisfies

$$n^{-1/2} \cdot \|\mathbf{V}_{r_k}^\top(\mathbf{y} - \widehat{\mathbf{y}}^{(T)})\|_2 \leqslant 2\left(1 - \Omega\left(k^{-d-1}\right)\right)^T \cdot n^{-1/2} \cdot \|\mathbf{V}_{r_k}^\top\mathbf{y}\|_2 + \epsilon,$$

where $r_k = \sum_{k'=0}^{k-1} N(d, k')$ and $\mathbf{V}_{r_k} = (n^{-1/2}\phi_j(\mathbf{x}_i))_{n \times r_k}$ with $\phi_1, \ldots, \phi_{r_k}$ being a set of orthonomal spherical harmonics of degrees up to $k - 1$.

**Corollary 3.8.** Suppose that $d \gg k$, and the sample $\{\mathbf{x}_i\}_{i=1}^n$ follows the uniform distribution $\tau_d$ on the unit sphere $\mathbb{S}^d$. For any $\epsilon, \delta > 0$ and integer $k$, if $n \geqslant \widetilde{\Omega}(\epsilon^{-2}d^{2k-2}r_k^2)$, $m \geqslant \widetilde{\Omega}(\mathrm{poly}(T, d^{k-2}, \epsilon^{-1}))$, then with probability at least $1 - \delta$, Algorithm 1 with $\eta = \widetilde{\mathcal{O}}((m\theta^2)^{-1})$, $\theta = \widetilde{\mathcal{O}}(\epsilon)$ satisfies

$$n^{-1/2} \cdot \|\mathbf{V}_{r_k}^\top(\mathbf{y} - \widehat{\mathbf{y}}^{(T)})\|_2 \leqslant 2\left(1 - \Omega\left(d^{-k+2}\right)\right)^T \cdot n^{-1/2} \cdot \|\mathbf{V}_{r_k}^\top\mathbf{y}\|_2 + \epsilon,$$

where $r_k = \sum_{k'=0}^{k-1} N(d, k')$ and $\mathbf{V}_{r_k} = (n^{-1/2}\phi_j(\mathbf{x}_i))_{n \times r_k}$ with $\phi_1, \ldots, \phi_{r_k}$ being a set of orthonomal spherical harmonics of degrees up to $k - 1$.

Corollaries 3.7 and 3.8 further illustrate the spectral bias of neural networks by providing exact calculations of $\lambda_{r_k}$, $\mathbf{V}_{r_k}$ and $M$ in Theorem 3.2. They show that if the input distribution is uniform over unit sphere, then spherical harmonics with lower degrees are learned first by over-parameterized neural networks.

**Remark 3.9.** In Corollaries 3.7 and 3.8, it shows that the conditions on $n$ and $m$ depend exponentially on either $k$ or $d$. We would like to emphasize that such exponential dependency is reasonable and unavoidable. In our case, we can take the $d \gg k$ setting as an example. The exponential dependency in $k$ is a natural consequence of the fact that in high dimensional space, there are a large number of linearly independent polynomials even for very low degrees. It is apparently only reasonable to expect to learn less than $n$ independent components of the true function, which means that it is unavoidable to assume

$$n \geqslant r_k = \sum_{k'=0}^{k-1} N(d, k') = \sum_{k'=0}^{k-1} \frac{2k' + d - 1}{k'}\binom{k'+d-2}{d-1} = \sum_{k'=0}^{k-1} \frac{2k' + d - 1}{k'}\binom{k'+d-2}{k'-1} = \Omega(d^{k-1}).$$

Similar arguments can apply to the requirement of $m$ and the $k \gg d$ setting.

## 4 EXPERIMENTS

In this section we illustrate our results by training neural networks on synthetic data. Across all tasks, we train a two-layer hidden neural networks with 4096 neurons and initialize it exactly as defined in the setup. The optimization method is vanilla full gradient descent. We sample 1000 training data which is uniformly sampled from the unit sphere in $\mathbb{R}^{10}$.

### 4.1 LEARNING COMBINATION OF SPHERICAL HARMONICS

First, we show a result when the target function is exactly linear combination of spherical harmonics. The target function is explicitly defined as

$$f^*(\mathbf{x}) = a_1 \cdot P_1(\langle \boldsymbol{\zeta}_1, \mathbf{x} \rangle) + a_2 \cdot P_2(\langle \boldsymbol{\zeta}_2, \mathbf{x} \rangle) + a_4 \cdot P_4(\langle \boldsymbol{\zeta}_4, \mathbf{x} \rangle),$$

where the $P_k(t)$ is the Gegenbauer polynomial, and $\boldsymbol{\zeta}_k$, $k = 1, 2, 4$ are fixed vectors that are independently generated from uniform distribution on unit sphere in $\mathbb{R}^{10}$ in our experiments. Note that according to the addition formula $\sum_{j=1}^{N(d,k)} Y_{k,j}(\mathbf{x}) Y_{k,j}(\mathbf{y}) = N(d,k) P_k(\langle \mathbf{x}, \mathbf{y} \rangle)$, every normalized Gegenbauer polynomial is a spherical harmonic, so $f^*(\mathbf{x})$ is a linear combination of spherical harmonics of order 1,2 and 4. The higher odd-order Gegenbauer polynomials are omitted because the spectral analysis showed that $\mu_k = 0$ for $k = 3, 5, 7 \ldots$.

Following the setting in section 3.1, we denote $\mathbf{v}_k = n^{-1/2}(P_k(\mathbf{x}_1), \ldots, P_k(\mathbf{x}_n))$. By Lemma 3.2 $\mathbf{v}_k$'s are almost orthonormal with high probability. So we can define the (approximate) projection length of residual $\mathbf{r}^{(t)}$ onto $\mathbf{v}_k$ at step $t$ as

$$\widehat{a}_k = |\mathbf{v}_k^\top \mathbf{r}^{(t)}|,$$

where $\mathbf{r}^{(t)} = (f^*(\mathbf{x}_1) - \theta f_{\mathbf{W}^{(t)}}(\mathbf{x}_1), \ldots, f^*(\mathbf{x}_n) - \theta f_{\mathbf{W}^{(t)}}(\mathbf{x}_n))$ and $f_{\mathbf{W}^{(t)}}(\mathbf{x})$ is the neural network function.

**Remark 4.1.** Here $\widehat{a}_k$ is the projection length onto an approximate vector. In the function space, we can also project the residual function $r(\mathbf{x}) = f^*(\mathbf{x}) - \theta f_{\mathbf{W}^{(t)}}(\mathbf{x})$ onto the orthonormal Gegenbauer functions $P_k(\mathbf{x})$. Replacing the training data with randomly sampled data points $\mathbf{x}_i$ can lead to a random estimate of the projection length in function space. We provide the corresponding result for freshly sampled points in Appendix E.1.

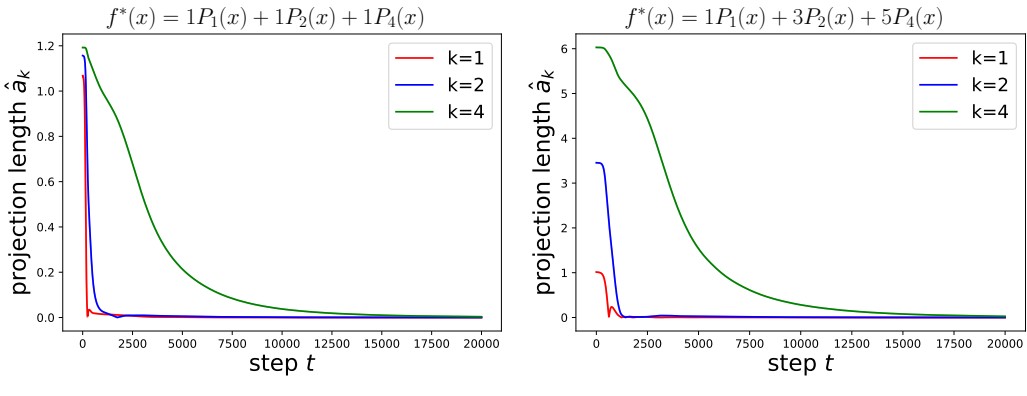

(a) components with the same scale  (b) components with different scale

Figure 1: Convergence curve for projection length onto different components. (a) shows the curve when the target function have different component with the same scale. (b) shows the curve when the higner-order components have larger scale. Both illustrate that the lower-order components are learned faster. Log-scale figures are shown in Appendix E.2.

The results are showned in Figure 1. It can be seen that at the beginning of training, the residual at the lowest frequency ($k = 1$) converges to zero first and then the second lowest ($k = 2$). The highest frequency component is the last one to converge. Following the setting of Rahaman et al. (2018) we assign high frequencies a larger scale, expecting that larger scale will introduce a better descending speed. Still, the low frequencies are regressed first.

## 4.2 LEARNING FUNCTIONS OF SIMPLE FORM

Apart from the synthesized low frequency function, we also showed the dynamics of normal functions' projection to $P_k(x)$. These functions, though in a simple form, have non-zero components in almost all frequencies. In this subsection we further show our results still apply when all frequencies exist in the target function, which is given by $f^*(\mathbf{x}) = \sum_i \cos(a_i \langle \boldsymbol{\zeta}, \mathbf{x} \rangle)$ or $f^*(\mathbf{x}) = \sum_i \langle \boldsymbol{\zeta}, \mathbf{x} \rangle^{p_i}$, where $\boldsymbol{\zeta}$ is a fixed unit vector. The coefficients of given components are calculated in the same way as in Section 4.1.

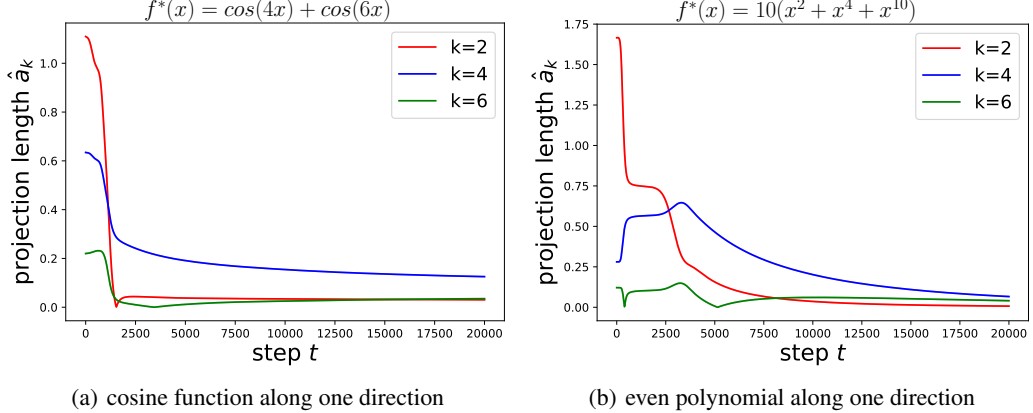

(a) cosine function along one direction      (b) even polynomial along one direction

Figure 2: Convergence curve for different components. (a) shows the curve of a trigonometric function. (b) shows the curve of a polynomial with even degrees. Both exhibits similar tendency as combination of spherical harmonics.

Figure 2 shows that even for arbitrarily chosen functions of simple form, the networks can still first learn the low frequency components of the target function. Notice that early in training not all the curves may descend, we believe this is due to the unseen components' influence on the gradient. Again, as the training proceeds, the convergence is controlled at the predicted rate.

**Remark 4.2.** The reason why we only use cosine function and even polynomial is that the only odd basis function with non-zero eigenvalue is $P_1(\mathbf{x})$. To show a general tendency it is better to restrict the target function in the even function space.

## 5 CONCLUSION AND DISCUSSION

In this paper, we give theoretical justification for spectral bias through a detailed analysis of the convergence behavior of two-layer neural networks with ReLU activation function. We show that the convergence of gradient descent in different directions depends on the corresponding eigenvalues and essentially exhibits different convergence rates. We show Mercer decomposition of neural tangent kernel and give explicit order of eigenvalues of integral operator with respect to the neural tangent kernel when the data is uniformly distributed on the unit sphere $\mathbb{S}^d$. Combined with the convergence analysis, we give the exact order of convergence rate on different directions. We also conduct experiments on synthetic data to support our theoretical result.

So far, we have considered the upper bound for convergence with respect to low frequency components and present comprehensive theorem to explain the spectral bias. One desired improvement is to give the lower bound of convergence with respect to high frequency components, which is essential to establish tighter characterization of spectral-biased optimization. It is also interesting to extend our result to other training algorithms like Adam, where the analysis in Wu et al. (2019); Zhou et al. (2018) might be implemented with a more careful quantification on the projection of residual along different directions. Another potential improvement is to generalize the result to multi-layer neural networks, which might require different techniques since our analysis heavily rely on exactly computing the eigenvalues of the neural tangent kernel. It is also an important direction to weaken the requirement on over-parameterization, or study the spectral bias in a non-NTK regime to furthur close the gap between theory and practice.

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

## A  REVIEW ON SPHERICAL HARMONICS

In this section, we give a brief review on relevant concepts in spherical harmonics. For more detials, see Bach (2017), Bietti and Mairal (2019), Frye and Efthimiou (2012) and Atkinson and Han (2012) for references.

We consider the unit sphere $\mathbb{S}^d = \{\mathbf{x} \in \mathbb{R}^{d+1} : \|\mathbf{x}\|_2 = 1\}$, whose surface area is given by $\omega_d = 2\pi^{(d+1)/2}/\Gamma((d+1)/2)$ and denote $\tau_d$ the uniform measure on the sphere.

For any $k \geqslant 1$, we consider a set of spherical harmonics $\left\{Y_{k,j} : \mathbb{S}^d \to \mathbb{R} | 1 \leqslant j \leqslant N(d,k) = \frac{2k+d-1}{k}\binom{k+d-2}{d-1}\right\}$. They form an orthonormal basis and satisfy the following equation $\langle Y_{ki}, Y_{sj}\rangle_{\mathbb{S}^d} = \int_{\mathbb{S}^d} Y_{ki}(x)Y_{sj}(x)d\tau_d(x) = \delta_{ij}\delta_{sk}$. Moreover, since they are homogeneous functions of degree $k$, it is clear that $Y_k(x)$ has the same parity as $k$.

We have the addition formula

$$\sum_{j=1}^{N(d,k)} Y_{k,j}(\mathbf{x})Y_{k,j}(\mathbf{y}) = N(d,k)P_k(\langle \mathbf{x}, \mathbf{y}\rangle), \tag{A.1}$$

where $P_k(t)$ is the Legendre polynomial of degree $k$ in $d+1$ dimensions, explicitly given by (Rodrigues' formula)

$$P_k(t) = \left(-\frac{1}{2}\right)^k \frac{\Gamma\left(\frac{d}{2}\right)}{\Gamma\left(k+\frac{d}{2}\right)}\left(1-t^2\right)^{\frac{2-d}{2}}\left(\frac{d}{dt}\right)^k \left(1-t^2\right)^{k+\frac{d-2}{2}}.$$

We can also see that $P_k(t)$, the Legendre polynomial of degree $k$ shares the same parity with $k$. By the orthogonality and the addition formula (A.1) we have,

$$\int_{\mathbb{S}^d} P_j(\langle \mathbf{w}, \mathbf{x}\rangle)P_k(\langle \mathbf{w}, \mathbf{y}\rangle)d\tau_d(\mathbf{w}) = \frac{\delta_{jk}}{N(d,k)}P_k(\langle \mathbf{x}, \mathbf{y}\rangle). \tag{A.2}$$

Further we have the recurrence relation for the Legendre polynomials,

$$tP_k(t) = \frac{k}{2k+d-1}P_{k-1}(t) + \frac{k+d-1}{2k+d-1}P_{k+1}(t), \tag{A.3}$$

for $k \geqslant 1$ and $tP_0(t) = P_1(t)$ for $k = 0$.

The Hecke-Funk formula is given for a spherical harmonic $Y_k$ of degree $k$

$$\int_{\mathbb{S}^d} f(\langle \mathbf{x}, \mathbf{y}\rangle)Y_k(\mathbf{y})dt'_d(\mathbf{y}) = \frac{\omega_{d-1}}{\omega_d}Y_k(\mathbf{x})\int_{-1}^{1} f(t)P_k(t)(1-t^2)^{(d-2)/2}dt.$$

## B  PROOF OF MAIN THEOREMS

### B.1  PROOF OF THEOREM 3.2

In this section we give the proof of Theorem 3.2. The core idea of our proof is to establish connections between neural network gradients throughout training and the neural tangent kernel. To do so, we first introduce the following definitions and notations.

Define $\mathbf{K}^{(0)} = m^{-1}(\langle \nabla_{\mathbf{W}} f_{\mathbf{W}^{(0)}}(\mathbf{x}_i), \nabla_{\mathbf{W}} f_{\mathbf{W}^{(0)}}(\mathbf{x}_j)\rangle)_{n \times n}$, $\mathbf{K}^{(\infty)} = (\kappa(\mathbf{x}_i, \mathbf{x}_j))_{n \times n} = \lim_{m\to\infty} \mathbf{K}^{(0)}$. Let $\{\hat{\lambda}_i\}_{i=1}^n$, $\hat{\lambda}_1 \geqslant \cdots \geqslant \hat{\lambda}_n$ be the eigenvalues of $n^{-1}\mathbf{K}^{\infty}$, and $\hat{\mathbf{v}}_1, \ldots, \hat{\mathbf{v}}_n$ be the corresponding eigenvectors. Set $\hat{\mathbf{V}}_{r_k} = (\hat{\mathbf{v}}_1, \ldots, \hat{\mathbf{v}}_{r_k})$, $\hat{\mathbf{V}}_{r_k}^{\perp} = (\hat{\mathbf{v}}_{r_k+1}, \ldots, \hat{\mathbf{v}}_n)$.

For notation simplicity, we denote $\nabla_{\mathbf{W}} f_{\mathbf{W}^{(0)}}(\mathbf{x}_i) = [\nabla_{\mathbf{W}} f_{\mathbf{W}}(\mathbf{x}_i)]\big|_{\mathbf{W}=\mathbf{W}^{(0)}}$, $\nabla_{\mathbf{W}_l} f_{\mathbf{W}^{(0)}}(\mathbf{x}_i) = [\nabla_{\mathbf{W}_l} f_{\mathbf{W}}(\mathbf{x}_i)]\big|_{\mathbf{W}=\mathbf{W}^{(0)}}$, $l = 1, 2$.

The following lemma is a direct application of Proposition 1 in Smale and Zhou (2009) or Proposition 10 in Rosasco et al. (2010). It bounds the difference between the eigenvalues of NTK and their finite-width counterparts.

**Lemma B.1.** For any $\delta > 0$, with probability at least $1 - \delta$,

$$|\lambda_i - \widehat{\lambda}_i| \leqslant \mathcal{O}(\sqrt{\log(1/\delta)/n}).$$

The following lemma is partly summarized from the proof of equation (44) in Su and Yang (2019). Its purpose is to further connect the eigenfunctions of NTK with their finite-width, finite-sample counterparts.

**Lemma B.2.** Suppose that $|\phi_i(\mathbf{x})| \leqslant M$ for all $\mathbf{x} \in S^d$. There exist absolute constants $C, C', c'' > 0$, such that for any $\delta > 0$ and integer $k$ with $r_k \leqslant n$, if $n \geqslant C(\lambda_{r_k} - \lambda_{r_k+1})^{-2} \log(1/\delta)$, then with probability at least $1 - \delta$,

$$\|\mathbf{V}_{r_k}^\top \widehat{\mathbf{V}}_{r_k}^\perp\|_F \leqslant C' \frac{1}{\lambda_{r_k} - \lambda_{r_k+1}} \cdot \sqrt{\frac{\log(1/\delta)}{n}},$$

$$\|\mathbf{V}_{r_k} \mathbf{V}_{r_k}^\top - \widehat{\mathbf{V}}_{r_k} \widehat{\mathbf{V}}_{r_k}^\top\|_2 \leqslant C''\left[\frac{1}{\lambda_{r_k} - \lambda_{r_k+1}} \cdot \sqrt{\frac{\log(1/\delta)}{n}} + M^2 r_k \sqrt{\frac{\log(r_k/\delta)}{n}}\right].$$

The following two lemmas are key to characterize the dynamics of the residual throughout training. These two results, especially Lemma B.4, are the ones that distinguish our analysis from previous works on neural network training in the NTK regime (Du et al., 2018b; Su and Yang, 2019), since our analysis provides more careful characterization on the residual along different directions.

**Lemma B.3.** Suppose that the iterates of gradient descent $\mathbf{W}^{(0)}, \ldots, \mathbf{W}^{(t)}$ are inside the ball $\mathcal{B}(\mathbf{W}^{(0)}, \omega)$. If $\omega \leqslant \mathcal{O}([\log(m)]^{-3/2})$, then with probability at least $1 - \mathcal{O}(n^2) \cdot \exp[-\Omega(m\omega^{2/3})]$,

$$\mathbf{y} - \widehat{\mathbf{y}}^{(t'+1)} = [\mathbf{I} - (\eta m \theta^2/n)\mathbf{K}^\infty](\mathbf{y} - \widehat{\mathbf{y}}^{(t')}) + \mathbf{e}^{(t)}, \quad \|\mathbf{e}^{(t')}\|_2 \leqslant \widetilde{\mathcal{O}}(\omega^{1/3}\eta m \theta^2) \cdot \|\mathbf{y} - \widehat{\mathbf{y}}^{(t')}\|_2$$

for all $t' = 0, \ldots, t-1$, where $\mathbf{y} = (y_1, \ldots, y_n)^\top$, $\widehat{\mathbf{y}}^{(t')} = \theta \cdot (f_{\mathbf{W}^{(t')}}(\mathbf{x}_1), \ldots, f_{\mathbf{W}^{(t')}}(\mathbf{x}_n))^\top$.

**Lemma B.4.** Suppose that the iterates of gradient descent $\mathbf{W}^{(0)}, \ldots, \mathbf{W}^{(t)}$ are inside the ball $\mathcal{B}(\mathbf{W}^{(0)}, \omega)$. If $\omega \leqslant \widetilde{\mathcal{O}}(\min\{[\log(m)]^{-3/2}, \lambda_{r_k}^3, (\eta m)^{-3}\})$ and $n \geqslant \widetilde{\mathcal{O}}(\lambda_{r_k}^{-2})$, then with probability at least $1 - \mathcal{O}(t^2 n^2) \cdot \exp[-\Omega(m\omega^{2/3})]$,

$$\|(\widehat{\mathbf{V}}_{r_k}^\perp)^\top(\mathbf{y} - \widehat{\mathbf{y}}^{(t')})\|_2 \leqslant \|(\widehat{\mathbf{V}}_{r_k}^\perp)^\top(\mathbf{y} - \widehat{\mathbf{y}}^{(0)})\|_2 + t' \cdot \omega^{1/3}\eta m \theta^2 \cdot \sqrt{n} \cdot \widetilde{\mathcal{O}}(1 + \omega\sqrt{m}) \tag{B.1}$$

$$\|\widehat{\mathbf{V}}_{r_k}^\top(\mathbf{y} - \widehat{\mathbf{y}}^{(t')})\|_2 \leqslant (1 - \eta m \theta^2 \lambda_{r_k}/2)^{t'}\|\widehat{\mathbf{V}}_{r_k}^\top(\mathbf{y} - \widehat{\mathbf{y}}^{(0)})\|_2 + t'\lambda_{r_k}^{-1} \cdot \omega^{2/3}\eta m \theta^2 \cdot \sqrt{n} \cdot \widetilde{\mathcal{O}}(1 + \omega\sqrt{m})$$
$$+ \lambda_{r_k}^{-1} \cdot \widetilde{\mathcal{O}}(\omega^{1/3}) \cdot \|(\widehat{\mathbf{V}}_{r_k}^\perp)^\top(\mathbf{y} - \widehat{\mathbf{y}}^{(0)})\|_2 \tag{B.2}$$

$$\|\mathbf{y} - \widehat{\mathbf{y}}^{(t')}\|_2 \leqslant \widetilde{\mathcal{O}}(\sqrt{n}) \cdot (1 - \eta m \theta^2 \lambda_{r_k}/2)^{t'} + \widetilde{\mathcal{O}}(\sqrt{n} \cdot (\eta m \theta^2 \lambda_{r_k})^{-1})$$
$$+ \lambda_{r_k}^{-1} t' \omega^{1/3} \cdot \sqrt{n} \cdot \widetilde{\mathcal{O}}(1 + \omega\sqrt{m}) \tag{B.3}$$

for all $t' = 0, \ldots, t-1$.

Now we are ready to prove Theorem 3.2.

*Proof of Theorem 3.2.* Define $\omega = \overline{C}T/(\lambda_{r_k}\sqrt{m})$ for some small enough absolute constant $\overline{C}$. Then by union bound, as long as $m \geqslant \widetilde{\Omega}(\text{poly}(T, \lambda_{r_k}^{-1}, \epsilon^{-1}))$, the conditions on $\omega$ given in Lemmas D.2, D.4, D.5, B.3 and B.4 are all satisfied.

We first show that all the iterates $\mathbf{W}^{(0)}, \ldots, \mathbf{W}^{(T)}$ are inside the ball $\mathcal{B}(\mathbf{W}^{(0)}, \omega)$. We prove this result by inductively show that $\mathbf{W}^{(t)} \in \mathcal{B}(\mathbf{W}^{(0)}, \omega)$, $t = 0, \ldots, T$. First of all, it is clear that $\mathbf{W}^{(0)} \in \mathcal{B}(\mathbf{W}^{(0)}, \omega)$. Suppose that $\mathbf{W}^{(0)}, \ldots, \mathbf{W}^{(t)} \in \mathcal{B}(\mathbf{W}^{(0)}, \omega)$. Then the results of Lemmas D.2, D.4,

D.5, B.3 and B.4 hold for $\mathbf{W}^{(0)}, \ldots, \mathbf{W}^{(t)}$. Denote $\mathbf{u}^{(t)} = \mathbf{y} - \widehat{\mathbf{y}}^{(t)}$, $t \in T$. Then we have

$$
\begin{aligned}
\|\mathbf{W}_l^{(t+1)} - \mathbf{W}_l^{(0)}\|_F &\leqslant \sum_{t'=0}^{t} \|\mathbf{W}_l^{(t'+1)} - \mathbf{W}_l^{(t')}\|_F \\
&= \eta \sum_{t'=0}^{t} \left\| \frac{1}{n} \sum_{i=1}^{n} (y_i - \theta \cdot f_{\mathbf{W}^{(t)}}(\mathbf{x}_i)) \cdot \theta \cdot \nabla_{\mathbf{W}_l} f_{\mathbf{W}^{(t)}}(\mathbf{x}_i) \right\|_F \\
&\leqslant \eta\theta \sum_{t'=0}^{t} \frac{1}{n} \sum_{i=1}^{n} |y_i - \theta \cdot f_{\mathbf{W}^{(t)}}(\mathbf{x}_i)| \cdot \|\nabla_{\mathbf{W}_l} f_{\mathbf{W}^{(t)}}(\mathbf{x}_i)\|_F \\
&\leqslant C_1 \eta\theta\sqrt{m} \sum_{t'=0}^{t} \frac{1}{n} \sum_{i=1}^{n} |y_i - \theta \cdot f_{\mathbf{W}^{(t)}}(\mathbf{x}_i)| \\
&\leqslant C_1 \eta\theta\sqrt{m/n} \sum_{t'=0}^{t} \|\mathbf{y} - \widehat{\mathbf{y}}^{(t')}\|_2,
\end{aligned}
$$

where the second inequality follows by Lemma D.4. By Lemma B.4, we have

$$
\sum_{t'=0}^{t} \|\mathbf{y} - \widehat{\mathbf{y}}^{(t')}\|_2 \leqslant \widetilde{\mathcal{O}}(\sqrt{n}/(\eta m\theta^2 \lambda_{r_k})) + \widetilde{\mathcal{O}}(T\sqrt{n}/(\eta m\theta^2 \lambda_{r_k})) + \lambda_{r_k}^{-1} T^2 \omega^{1/3} \sqrt{n} \cdot \widetilde{\mathcal{O}}(1 + \omega\sqrt{m}).
$$

It then follows by the choice $\omega = \overline{C}T/(\theta\lambda_{r_k}\sqrt{m})$, $\eta = \widetilde{\mathcal{O}}((m\theta^2\lambda_{r_k})^{-1})$, $\theta = \widetilde{\mathcal{O}}(\epsilon)$ and the assumption $m \geqslant \widetilde{\mathcal{O}}(\text{poly}(T, \lambda_{r_k}^{-1}, \epsilon^{-1}))$ that $\|\mathbf{W}_l^{(t+1)} - \mathbf{W}_l^{(0)}\|_F \leqslant \omega$, $l = 1, 2$. Therefore by induction, we see that with probability at least $1 - \mathcal{O}(T^3 n^2) \cdot \exp[-\Omega(m\omega^{2/3})]$, $\mathbf{W}(0), \ldots, \mathbf{W}(T) \in \mathcal{B}(\mathbf{W}^{(0)}, \omega)$.

Applying Lemma B.4 then gives

$$
\begin{aligned}
n^{-1/2} \cdot \|\widehat{\mathbf{V}}_{r_k}^\top (\mathbf{y} - \widehat{\mathbf{y}}^{(T)})\|_2 \leqslant &(1 - \eta m\theta^2 \lambda_{r_k}/2)^T \cdot n^{-1/2} \cdot \|\widehat{\mathbf{V}}_{r_k}^\top (\mathbf{y} - \widehat{\mathbf{y}}^{(0)})\|_2 \\
&+ T\lambda_{r_k}^{-1} \cdot \omega^{2/3} \eta m\theta^2 \cdot \widetilde{\mathcal{O}}(1 + \omega\sqrt{m}) \\
&+ \lambda_{r_k}^{-1} \cdot \widetilde{\mathcal{O}}(\omega^{1/3}) \cdot n^{-1/2} \cdot \|(\widehat{\mathbf{V}}_{r_k}^\perp)^\top (\mathbf{y} - \widehat{\mathbf{y}}^{(0)})\|_2.
\end{aligned}
$$

Now by $\omega = \overline{C}T/(\lambda_{r_k}\sqrt{m})$, $\eta = \widetilde{\mathcal{O}}(\theta^2 m)^{-1}$ and the assumption that $m \geqslant m^* = \widetilde{\mathcal{O}}(\lambda_{r_k}^{-14} \cdot \epsilon^{-6})$, we obtain

$$
n^{-1/2} \cdot \|\widehat{\mathbf{V}}_{r_k}^\top (\mathbf{y} - \widehat{\mathbf{y}}^{(T)})\|_2 \leqslant (1 - \lambda_{r_k})^T \cdot n^{-1/2} \cdot \|\widehat{\mathbf{V}}_{r_k}^\top (\mathbf{y} - \widehat{\mathbf{y}}^{(0)})\|_2 + \epsilon/16. \tag{B.4}
$$

By Lemma 3.1, $\theta = \widetilde{\mathcal{O}}(\epsilon)$ and the assumptions $n \geqslant \widetilde{\Omega}(\max\{\epsilon^{-1}(\lambda_{r_k} - \lambda_{r_k+1})^{-1}, \epsilon^{-2}M^2 r_k^2\})$, the eigenvalues of $\mathbf{V}_{r_k}^\top \mathbf{V}_{r_k}$ are all between $1/\sqrt{2}$ and $\sqrt{2}$. Therefore by Lemma B.2 we have

$$
\begin{aligned}
\|\widehat{\mathbf{V}}_{r_k}^\top (\mathbf{y} - \widehat{\mathbf{y}}^{(T)})\|_2 &= \|\widehat{\mathbf{V}}_{r_k} \widehat{\mathbf{V}}_{r_k}^\top (\mathbf{y} - \widehat{\mathbf{y}}^{(T)})\|_2 \\
&\geqslant \|\mathbf{V}_{r_k} \mathbf{V}_{r_k}^\top (\mathbf{y} - \widehat{\mathbf{y}}^{(T)})\|_2 - \|(\mathbf{V}_{r_k} \mathbf{V}_{r_k}^\top - \widehat{\mathbf{V}}_{r_k} \widehat{\mathbf{V}}_{r_k}^\top)(\mathbf{y} - \widehat{\mathbf{y}}^{(T)})\|_2 \\
&\geqslant \|\mathbf{V}_{r_k}^\top (\mathbf{y} - \widehat{\mathbf{y}}^{(T)})\|_2/\sqrt{2} - \sqrt{n} \cdot \mathcal{O}\left(\frac{1}{\lambda_{r_k} - \lambda_{r_k+1}} \cdot \sqrt{\frac{\log(1/\delta)}{n}} + M r_k \sqrt{\frac{\log(r_k/\delta)}{n}}\right) \\
&\geqslant \|\mathbf{V}_{r_k}^\top (\mathbf{y} - \widehat{\mathbf{y}}^{(T)})\|_2/\sqrt{2} - \epsilon\sqrt{n}/16.
\end{aligned}
$$

Similarly,

$$
\begin{aligned}
\|\widehat{\mathbf{V}}_{r_k}^\top (\mathbf{y} - \widehat{\mathbf{y}}^{(0)})\|_2 &\leqslant \sqrt{2} \cdot \|\mathbf{V}_{r_k}^\top (\mathbf{y} - \widehat{\mathbf{y}}^{(0)})\|_2 + \epsilon\sqrt{n}/16 \\
&\leqslant \sqrt{2} \cdot \|\mathbf{V}_{r_k}^\top \mathbf{y}\|_2 + \sqrt{2} \cdot \|\mathbf{V}_{r_k}^\top \widehat{\mathbf{y}}^{(0)}\|_2 + \epsilon\sqrt{n}/16.
\end{aligned}
$$

By Lemma 3.1, with probability at least $1 - \delta$, $\|\mathbf{V}_{r_k}^\top\|_2 \leqslant 1 + C r_k M^2 \sqrt{\log(r_k/\delta)/n}$. Combining this result with Lemma D.3 gives $\|\mathbf{V}_{r_k}^\top \widehat{\mathbf{y}}^{(0)}\|_2 \leqslant \theta\mathcal{O}(\sqrt{n\log(n/\delta)}) \leqslant \epsilon\sqrt{n}/8$. Plugging the above estimates into (B.4) gives

$$
n^{-1/2} \cdot \|\mathbf{V}_{r_k}^\top (\mathbf{y} - \widehat{\mathbf{y}}^{(T)})\|_2 \leqslant 2(1 - \lambda_{r_k})^T \cdot n^{-1/2} \cdot \|\mathbf{V}_{r_k}^\top \mathbf{y}\|_2 + \epsilon.
$$

Applying union bounds completes the proof. $\qquad\square$

### B.2 PROOF OF THE THEOREM 3.4

*Proof of the Theorem 3.4.* The idea of the proof is close to that of Proposition 5 in (Bietti and Mairal, 2019) where they consider $k \gg d$ and we present a more general case including $k \gg d$ and $d \gg k$. For any function $g : \mathbb{S}^d \to \mathbb{R}$, by denoting $g_0(\mathbf{x}) = \int_{\mathbb{S}^d} g(\mathbf{y}) d\tau_d(\mathbf{y})$, it can be decomposed as

$$g(\mathbf{x}) = \sum_{k=0}^{\infty} g_k(\mathbf{x}) = \sum_{k=0}^{\infty} \sum_{j=1}^{N(d,k)} \int_{\mathbb{S}^d} Y_{kj}(\mathbf{y}) Y_{kj}(\mathbf{x}) g(\mathbf{y}) d\tau_d(\mathbf{y})$$

$$= \sum_{k=0}^{\infty} N(d,k) \int_{\mathbb{S}^d} g(\mathbf{y}) P_k(\langle \mathbf{x}, \mathbf{y} \rangle) d\tau_d(\mathbf{y}), \tag{B.5}$$

where we project function $g$ to spherical harmonics in the second equality and apply the addition equation in the last equality.

For a positive-definite dot-product kernel $\kappa(\mathbf{x}, \mathbf{x}') : \mathbb{S}^d \times \mathbb{S}^d \to \mathbb{R}$ which has the form $\kappa(\mathbf{x}, \mathbf{x}') = \widehat{\kappa}(\langle \mathbf{x}, \mathbf{x}' \rangle)$ for $\widehat{\kappa} : [-1, 1] \to \mathbb{R}$, we can present a decomposition by (B.5) if we consider $g(\mathbf{x}) = \phi(\langle \mathbf{x}, \mathbf{z} \rangle)$ for $\mathbf{z} \in \mathbb{S}^d$ and $\phi : [-1, 1] \to \mathbb{R}$,

$$\kappa(\mathbf{x}, \mathbf{x}') = \sum_{k=0}^{\infty} N(d,k) \int_{\mathbb{S}^d} \widehat{\kappa}(\langle \mathbf{y}, \mathbf{x}' \rangle) P_k(\langle \mathbf{y}, \mathbf{x} \rangle) d\tau_d(\mathbf{y})$$

$$= \sum_{k=0}^{\infty} N(d,k) \frac{\omega_{d-1}}{\omega_d} P_k(\langle \mathbf{x}, \mathbf{x}' \rangle) \int_{-1}^{1} \widehat{\kappa}(t) P_k(t) (1 - t^2)^{(d-2)/2} dt,$$

where we apply the Hecke-Funk formula and addition formula. By denoting $\lambda_k = (\omega_{d-1}/\omega_d) \int_{-1}^{1} \widehat{\kappa}(t) P_k(t) (1 - t^2)^{(d-2)/2} dt$ and the addition formula, we have

$$\kappa(\mathbf{x}, \mathbf{x}') = \sum_{k=0}^{\infty} \mu_k N(d,k) P_k(\langle \mathbf{x}, \mathbf{x}' \rangle) = \sum_{k=0}^{\infty} \mu_k \sum_{j=1}^{N(p,k)} Y_{k,j}(\mathbf{x}) Y_{k,j}(\mathbf{x}'). \tag{B.6}$$

This formula (B.6) is the Mercer decomposition for the kernel function $\kappa(\mathbf{x}, \mathbf{x}')$ and $\mu_k$ is exactly the eigenvalue of the integral operator $L_K$ on $L_2(\mathbb{S}^d)$ defined by

$$L_\kappa(f)(\mathbf{y}) = \int_{\mathbb{S}^d} \kappa(\mathbf{x}, \mathbf{y}) f(\mathbf{x}) d\tau_d(\mathbf{x}), \quad f \in L_2(\mathbb{S}^d).$$

By using same technique as $\kappa(\mathbf{x}, \mathbf{x}')$, we can derive a similar expression for $\sigma(\langle \mathbf{w}, \mathbf{x} \rangle) = \max\{\langle \mathbf{w}, \mathbf{x} \rangle, 0\}$ and $\sigma'(\langle \mathbf{w}, \mathbf{x} \rangle) = \mathbb{1}\{\langle \mathbf{w}, \mathbf{x} \rangle > 0\}$, since they are essentially dot-product function on $L_2(\mathbb{S}^d)$. We deliver the expression below without presenting proofs.

$$\sigma'(\langle \mathbf{w}, \mathbf{x} \rangle) = \sum_{k=0}^{\infty} \beta_{1,k} N(d,k) P_k(\langle \mathbf{w}, \mathbf{x} \rangle), \tag{B.7}$$

$$\sigma(\langle \mathbf{w}, \mathbf{x} \rangle) = \sum_{k=0}^{\infty} \beta_{2,k} N(d,k) P_k(\langle \mathbf{w}, \mathbf{x} \rangle), \tag{B.8}$$

where $\beta_{1,k} = (\omega_{d-1}/\omega_d) \int_{-1}^{1} \sigma(t) P_k(t) (1 - t^2)^{(d-2)/2} dt$ and $\beta_{2,k} = (\omega_{d-1}/\omega_d) \int_{-1}^{1} \sigma'(t) P_k(t) (1 - t^2)^{(d-2)/2} dt$. We add more comments on the values of $\beta_{1,k}$ and $\beta_{2,k}$. It has been pointed out in Bach (2017) that when $k > \alpha$ and when $k$ and $\alpha$ have same parity, we have $\beta_{\alpha+1,k} = 0$. This is because the Legendre polynomial $P_k(t)$ is orthogonal to any other polynomials of degree less than $k$ with respect to the density function $p(t) = (1 - t^2)^{(d-2)/2}$. Then we clearly know that $\beta_{1,k} = 0$ for $k = 2j$ and $\beta_{2,k} = 0$ for $k = 2j + 1$ with $j \in \mathbb{N}^+$.

For two kernel function defined in (2.2), we have

$$\kappa_1(\mathbf{x}, \mathbf{x}') = \mathbb{E}_{\mathbf{w} \sim N(\mathbf{0}, \mathbf{I})} \left[ \sigma'(\langle \mathbf{w}, \mathbf{x} \rangle) \sigma'(\langle \mathbf{w}, \mathbf{x}' \rangle) \right]$$

$$= \mathbb{E}_{\mathbf{w} \sim N(\mathbf{0}, \mathbf{I})} \left[ \sigma'(\langle \mathbf{w}/\|\mathbf{w}\|, \mathbf{x} \rangle) \sigma'(\langle \mathbf{w}/\|\mathbf{w}\|, \mathbf{x}' \rangle) \right]$$

$$= \int_{\mathbb{S}^d} \sigma'(\langle \mathbf{v}, \mathbf{x} \rangle) \sigma'(\langle \mathbf{v}, \mathbf{x}' \rangle) d\tau_d(\mathbf{v}). \tag{B.9}$$

The first equality holds because $\sigma'$ is 0-homogeneous function and the second equality is true since the normalized direction of a multivariate Gaussian random variable satisfies uniform distribution on the unit sphere. Similarly we can derive

$$\kappa_2(\mathbf{x}, \mathbf{x}') = (d+1) \int_{\mathbb{S}^d} \sigma(\langle \mathbf{v}, \mathbf{x} \rangle) \sigma(\langle \mathbf{v}, \mathbf{x}' \rangle) d\tau_d(\mathbf{v}). \tag{B.10}$$

By combing (A.2), (B.7), (B.8), (B.9) and (B.10), we can get

$$\kappa_1(\mathbf{x}, \mathbf{x}') = \sum_{k=0}^{\infty} \beta_{1,k}^2 N(d,k) P_k(\langle \mathbf{x}, \mathbf{x}' \rangle), \tag{B.11}$$

and

$$\kappa_2(\mathbf{x}, \mathbf{x}') = (d+1) \sum_{k=0}^{\infty} \beta_{2,k}^2 N(d,k) P_k(\langle \mathbf{x}, \mathbf{x}' \rangle). \tag{B.12}$$

Comparing (B.6), (B.11) and (B.12), we can easily show that

$$\mu_{1,k} = \beta_{1,k}^2 \text{ and } \mu_{2,k} = (d+1)\beta_{2,k}^2. \tag{B.13}$$

In Bach (2017), explicit expressions for $\beta_{1,k}$ and $\beta_{2,k}$ for $k \geqslant \alpha + 1$ are presented by

$$\beta_{\alpha+1,k} = \frac{d-1}{2\pi} \frac{\alpha!(-1)^{(k-1-\alpha)/2}}{2^k} \frac{\Gamma(d/2)\Gamma(k-\alpha)}{\Gamma(\frac{k-\alpha+1}{2})\Gamma(\frac{k+d+\alpha+1}{2})}.$$

By Stirling formula $\Gamma(x) \approx x^{x-1/2} e^{-x} \sqrt{2\pi}$, we have following expression of $\beta_{\alpha+1,k}$ for $k \geqslant \alpha+1$

$$\beta_{\alpha+1,k} = C(\alpha) \frac{(d-1)d^{\frac{d-1}{2}}(k-\alpha)^{k-\alpha-\frac{1}{2}}}{(k-\alpha+1)^{\frac{k-\alpha}{2}}(k+d+\alpha+1)^{\frac{k+d+\alpha}{2}}} = \Omega\left(d^{\frac{d+1}{2}} k^{\frac{k-\alpha-1}{2}}(k+d)^{\frac{-k-d-\alpha}{2}}\right)$$

where $C(\alpha) = \frac{\sqrt{2}\alpha!}{2\pi}\exp\{\alpha + 1\}$. Also $\beta_{\alpha+1,0} = \frac{d-1}{4\pi}\frac{\Gamma(\frac{\alpha+1}{2})\Gamma(\frac{d}{2})}{\Gamma(\frac{d+\alpha+2}{2})}$, $\beta_{1,1} = \frac{d-1}{2d\pi}$ and $\beta_{2,1} = \frac{d-1}{4\pi d}\frac{\Gamma(\frac{1}{2})\Gamma(\frac{d+2}{2})}{\Gamma(\frac{d+3}{2})}$. Thus combine (B.13) we know that $\mu_{\alpha+1,k} = \Omega\left(d^{d+1+\alpha} k^{k-\alpha-1}(k+d)^{-k-d-\alpha}\right)$.

By considering (A.3) and (B.6), we have

$$\mu_0 = \mu_{1,1} + 2\mu_{2,0},$$

$$\mu_{k'} = 0, \quad k' = 2j+1, \ j \in \mathbb{N}^+,$$

and

$$\mu_k = \frac{k}{2k+d-1}\mu_{1,k-1} + \frac{k+d-1}{2k+d-1}\mu_{1,k+1} + 2\mu_{2,k},$$

for $k \geqslant 1$ and $k \neq k'$. From the discussion above, we thus know exactly that for $k \geqslant 1$

$$\mu_k = \Omega\left(\max\left\{d^{d+1}k^{k-1}(k+d)^{-k-d}, d^{d+1}k^k(k+d)^{-k-d-1}, d^{d+2}k^{k-2}(k+d)^{-k-d-1}\right\}\right).$$

This finishes the proof. □

### B.3 Proof of the Corollaries 3.7 and 3.8

*Proof of the Corollaries 3.7 and 3.8.* We only need to bound $|\phi_j(\mathbf{x})|$ for $j \in [r_k]$ to finish the proof. Since now we assume input data follows uniform distribution on the unit sphere $\mathbb{S}^d$, $\phi_j(\mathbf{x})$ would be spherical harmonics of order at most $k$ for $j \in [r_k]$. For any spherical harmonics $Y_k$ of order $k$ and any point on $\mathbb{S}^d$, we have an upper bound (Proposition 4.16 in Frye and Efthimiou (2012))

$$|Y_k(\mathbf{x})| \leqslant \left(N(d,k) \int_{\mathbb{S}^d} Y_k^2(\mathbf{y}) d\tau_d(\mathbf{y})\right)^{\frac{1}{2}}.$$

Thus we know that $|\phi_j(\mathbf{x})| \leqslant \sqrt{N(d,k)}$. For $k \gg d$, we have $N(d,k) = \frac{2k+d-1}{k}\binom{k+d-2}{d-1} = \mathcal{O}(k^{d-1})$. For $d \gg k$, we have $N(d,k) = \frac{2k+d-1}{k}\binom{k+d-2}{d-1} = \mathcal{O}(d^k)$. This completes the proof. □

## C    PROOF OF TECHNICAL LEMMAS

### C.1    PROOF OF LEMMA B.2

*Proof of Lemma B.2.* The first inequality directly follows by equation (44) in Su and Yang (2019). To prove the second bound, we write $\mathbf{V}_{r_k} = \widehat{\mathbf{V}}_{r_k}\mathbf{A} + \widehat{\mathbf{V}}_{r_k}^{\perp}\mathbf{B}$, where $\mathbf{A} \in \mathbb{R}^{r_k \times r_k}$, $\mathbf{B} \in \mathbb{R}^{(n-r_k) \times r_k}$. Let $\xi_1 = C'(\lambda_{r_k} - \lambda_{r_k+1})^{-1} \cdot \sqrt{\log(1/\delta)/n}$, $\xi_2 = C'''M^2\sqrt{\log(r_k/\delta)/n}$ be the bounds given in the first inequality and Lemma 3.1. By the first inequality, we have

$$\|\mathbf{B}\|_F = \|\mathbf{B}^\top\|_F = \|\mathbf{V}_{r_k}^\top \widehat{\mathbf{V}}_{r_k}^{\perp}\|_F \le \xi_1.$$

Moreover, since $\mathbf{V}_{r_k}^\top \mathbf{V}_{r_k} = \mathbf{A}^\top\mathbf{A} + \mathbf{B}^\top\mathbf{B}$, by Lemma 3.1 we have

$$\|\mathbf{A}\mathbf{A}^\top - \mathbf{I}\|_2 = \|\mathbf{A}^\top\mathbf{A} - \mathbf{I}\|_2 \le \|\mathbf{V}_{r_k}^\top\mathbf{V}_{r_k} - \mathbf{I}\|_2 + \|\mathbf{B}^\top\mathbf{B}\|_2 \le r_k\xi_2 + \xi_1^2.$$

Therefore

$$
\begin{aligned}
\|\mathbf{V}_{r_k}&\mathbf{V}_{r_k}^\top - \widehat{\mathbf{V}}_{r_k}\widehat{\mathbf{V}}_{r_k}^\top\|_2 \\
&= \|\widehat{\mathbf{V}}_{r_k}\mathbf{A}\mathbf{A}^\top\widehat{\mathbf{V}}_{r_k}^\top + \widehat{\mathbf{V}}_{r_k}\mathbf{A}\mathbf{B}^\top(\widehat{\mathbf{V}}_{r_k}^{\perp})^\top + \widehat{\mathbf{V}}_{r_k}^{\perp}\mathbf{B}\mathbf{A}^\top\widehat{\mathbf{V}}_{r_k}^\top + \widehat{\mathbf{V}}_{r_k}^{\perp}\mathbf{B}\mathbf{B}^\top(\widehat{\mathbf{V}}_{r_k}^{\perp})^\top - \widehat{\mathbf{V}}_{r_k}\widehat{\mathbf{V}}_{r_k}^\top\|_2 \\
&\le \|\widehat{\mathbf{V}}_{r_k}(\mathbf{A}\mathbf{A}^\top - \mathbf{I})\widehat{\mathbf{V}}_{r_k}^\top\|_2 + \mathcal{O}(\|\mathbf{B}\|_2) \\
&= \|\mathbf{A}\mathbf{A}^\top - \mathbf{I}\|_2 + \mathcal{O}(\|\mathbf{B}\|_2) \\
&\le \mathcal{O}(r_k\xi_2 + \xi_1)
\end{aligned}
$$

Plugging in the definition of $\xi_1$ and $\xi_2$ completes the proof.    ☐

### C.2    PROOF OF LEMMA B.3

*Proof of Lemma B.3.* The gradient descent update formula gives

$$\mathbf{W}^{(t+1)} = \mathbf{W}^{(t)} + \frac{2\eta}{n}\sum_{i=1}^{n}(y_i - \theta f_{\mathbf{W}^{(t)}}(\mathbf{x}_i)) \cdot \theta\nabla_{\mathbf{W}}f_{\mathbf{W}^{(t)}}(\mathbf{x}_i). \tag{C.1}$$

For any $j \in [n]$, subtracting $\mathbf{W}^{(t)}$ and applying inner product with $\theta\nabla_{\mathbf{W}}f_{\mathbf{W}^{(t)}}(\mathbf{x}_j)$ on both sides gives

$$\theta\langle\nabla_{\mathbf{W}}f_{\mathbf{W}^{(t)}}(\mathbf{x}_j), \mathbf{W}^{(t+1)} - \mathbf{W}^{(t)}\rangle = \frac{2\eta\theta^2}{n}\sum_{i=1}^{n}(y_i - \widehat{y}_i^{(t)}) \cdot \langle\nabla_{\mathbf{W}}f_{\mathbf{W}^{(t)}}(\mathbf{x}_j), \nabla_{\mathbf{W}}f_{\mathbf{W}^{(t)}}(\mathbf{x}_i)\rangle.$$

Further rearranging terms then gives

$$y_j - (\widehat{\mathbf{y}}^{(t+1)})_j = y_j - (\widehat{\mathbf{y}}^{(t)})_j - \frac{2\eta m\theta^2}{n}\sum_{i=1}^{n}(y_i - \theta f_{\mathbf{W}^{(t)}}(\mathbf{x}_i)) \cdot \mathbf{K}_{i,j}^{\infty} + I_{1,j,t} + I_{2,j,t} + I_{3,j,t}, \tag{C.2}$$

where

$$
\begin{aligned}
I_{1,j,t} &= -\frac{2\eta\theta^2}{n}\sum_{i=1}^{n}(y_i - \theta f_{\mathbf{W}^{(t)}}(\mathbf{x}_i)) \cdot [\langle\nabla_{\mathbf{W}}f_{\mathbf{W}^{(t)}}(\mathbf{x}_j), \nabla_{\mathbf{W}}f_{\mathbf{W}^{(t)}}(\mathbf{x}_i)\rangle - m\mathbf{K}_{i,j}^{(0)}], \\
I_{2,j,t} &= -\frac{2\eta m\theta^2}{n}\sum_{i=1}^{n}(y_i - \theta f_{\mathbf{W}^{(t)}}(\mathbf{x}_i)) \cdot (\mathbf{K}_{i,j}^{(0)} - \mathbf{K}_{i,j}^{\infty}), \\
I_{3,j,t} &= -\theta \cdot [f_{\mathbf{W}^{(t+1)}}(\mathbf{x}_j) - f_{\mathbf{W}^{(t)}}(\mathbf{x}_j) - \langle\nabla_{\mathbf{W}}f_{\mathbf{W}^{(t)}}(\mathbf{x}_j), \mathbf{W}^{(t+1)} - \mathbf{W}^{(t)}\rangle].
\end{aligned}
$$

For $I_{1,j,t}$, by Lemma D.6, we have

$$|I_{1,j,t}| \le \mathcal{O}(\omega^{1/3}\eta m\theta^2) \cdot \frac{1}{n}\sum_{i=1}^{n}|y_i - \theta f_{\mathbf{W}^{(t)}}(\mathbf{x}_i)| \le \mathcal{O}(\omega^{1/3}\eta m\theta^2) \cdot \|\mathbf{y} - \widehat{\mathbf{y}}^{(t)}\|_2/\sqrt{n}$$

with probability at least $1 - \mathcal{O}(n) \cdot \exp[-\Omega(m\omega^{2/3})]$. For $I_{2,j,t}$, by Bernstein inequality and union bound, with probability at least $1 - \mathcal{O}(n^2) \cdot \exp(-\Omega(m\omega^{2/3}))$, we have

$$\left| \mathbf{K}_{i,j}^\infty - \mathbf{K}_{i,j}^{(0)} \right| \leqslant \mathcal{O}(\omega^{1/3})$$

for all $i, j \in [n]$. Therefore

$$|I_{2,j,t}| \leqslant \mathcal{O}(\omega^{1/3}\eta m\theta^2) \cdot \frac{1}{n} \sum_{i=1}^n |y_i - \theta f_{\mathbf{W}^{(t)}}(\mathbf{x}_i)| \leqslant \mathcal{O}(\omega^{1/3}\eta m\theta^2) \cdot \|\mathbf{y} - \widehat{\mathbf{y}}^{(t)}\|_2/\sqrt{n}.$$

For $I_{3,j,t}$, we have

$$\begin{aligned}
I_{3,j,t} &\leqslant \widetilde{\mathcal{O}}(\omega^{1/3}\sqrt{m}\theta) \cdot \|\mathbf{W}_1^{(t+1)} - \mathbf{W}_1^{(t)}\|_2 \\
&\leqslant \widetilde{\mathcal{O}}(\omega^{1/3}\sqrt{m}\theta) \cdot \frac{2\eta}{n} \sum_{i=1}^n |y_i - \theta f_{\mathbf{W}^{(t)}}(\mathbf{x}_i)| \cdot \theta \cdot \|\nabla_{\mathbf{W}_1} f_{\mathbf{W}^{(t)}}(\mathbf{x}_i)\|_2 \\
&\leqslant \widetilde{\mathcal{O}}(\omega^{1/3}\eta m\theta^2) \cdot \frac{1}{n} \sum_{i=1}^n |y_i - \theta f_{\mathbf{W}^{(t)}}(\mathbf{x}_i)| \\
&\leqslant \widetilde{\mathcal{O}}(\omega^{1/3}\eta m\theta^2) \cdot \|\mathbf{y} - \widehat{\mathbf{y}}^{(t)}\|_2/\sqrt{n},
\end{aligned}$$

where the first inequality follows by Lemmas D.2, the second inequality is obtained from (C.1), and the third inequality follows by Lemma D.4. Setting the $j$-th entry of $\mathbf{e}^{(t)}$ as $I_{1,j,t} + I_{2,j,t} + I_{3,j,t}$ and writing (C.2) into matrix form completes the proof. $\qquad\square$

## C.3    PROOF OF LEMMA B.4

*Proof of Lemma B.4.* Denote $\mathbf{u}^{(t)} = \mathbf{y} - \widehat{\mathbf{y}}^{(t)}$, $t \in T$. Then we have

$$\begin{aligned}
\|(\widehat{\mathbf{V}}_{r_k}^\perp)^\top \mathbf{u}^{(t'+1)}\|_2 &\leqslant \|(\widehat{\mathbf{V}}_{r_k}^\perp)^\top [\mathbf{I} - (\eta m\theta^2/n)\mathbf{K}^\infty]\mathbf{u}^{(t')}\|_2 + \widetilde{\mathcal{O}}(\omega^{1/3}\eta m\theta^2) \cdot \|\mathbf{u}^{(t')}\|_2 \\
&\leqslant \|(\widehat{\mathbf{V}}_{r_k}^\perp)^\top \mathbf{u}^{(t')}\|_2 + \widetilde{\mathcal{O}}(\omega^{1/3}\eta m\theta^2) \cdot \sqrt{n} \cdot \widetilde{\mathcal{O}}(1 + \omega\sqrt{m}),
\end{aligned}$$

where the first inequality follows by Lemma B.3, and the second inequality follows by Lemma D.5. Therefore we have

$$\|(\widehat{\mathbf{V}}_{r_k}^\perp)^\top \mathbf{u}^{(t')}\|_2 \leqslant \|(\widehat{\mathbf{V}}_{r_k}^\perp)^\top \mathbf{u}^{(0)}\|_2 + t' \cdot \omega^{1/3}\eta m\theta^2 \cdot \sqrt{n} \cdot \widetilde{\mathcal{O}}(1 + \omega\sqrt{m}),$$

for $t' = 0, \ldots, t$. This completes the proof of (B.1). Similarly, we have

$$\begin{aligned}
\|\widehat{\mathbf{V}}_{r_k}^\top \mathbf{u}^{(t'+1)}\|_2 &\leqslant \|\widehat{\mathbf{V}}_{r_k}^\top [\mathbf{I} - (\eta m\theta^2/n)\mathbf{K}^\infty]\mathbf{u}^{(t')}\|_2 + \widetilde{\mathcal{O}}(\omega^{1/3}\eta m\theta^2) \cdot \|\mathbf{u}^{(t')}\|_2 \\
&\leqslant (1 - \eta m\theta^2\widehat{\lambda}_{r_k})\|\widehat{\mathbf{V}}_{r_k}^\top \mathbf{u}^{(t')}\|_2 + \widetilde{\mathcal{O}}(\omega^{1/3}\eta m\theta^2) \cdot (\|\widehat{\mathbf{V}}_{r_k}^\top \mathbf{u}^{(t')}\|_2 + \|(\widehat{\mathbf{V}}_{r_k}^\perp)^\top \mathbf{u}^{(t')}\|_2) \\
&\leqslant (1 - \eta m\theta^2\lambda_{r_k}/2)\|\widehat{\mathbf{V}}_{r_k}^\top \mathbf{u}^{(t')}\|_2 + \widetilde{\mathcal{O}}(\omega^{1/3}\eta m\theta^2) \cdot \|(\widehat{\mathbf{V}}_{r_k}^\perp)^\top \mathbf{u}^{(t')}\|_2 \\
&\leqslant (1 - \eta m\theta^2\lambda_{r_k}/2)\|\widehat{\mathbf{V}}_{r_k}^\top \mathbf{u}^{(t')}\|_2 + t' \cdot (\omega^{1/3}\eta m\theta^2)^2 \cdot \sqrt{n} \cdot \widetilde{\mathcal{O}}(1 + \omega\sqrt{m}) \\
&\quad + \widetilde{\mathcal{O}}(\omega^{1/3}\eta m\theta^2) \cdot \|(\widehat{\mathbf{V}}_{r_k}^\perp)^\top \mathbf{u}^{(0)}\|_2
\end{aligned}$$

for $t' = 0, \ldots, t-1$, where the third inequality is by Lemma B.1 and the assumption that $\omega \leqslant \widetilde{\mathcal{O}}(\lambda_{r_k}^3)$, $n \geqslant \widetilde{\mathcal{O}}(\lambda_{r_k}^{-2})$, and the fourth inequality is by (B.1). Therefore we have

$$\begin{aligned}
\|\widehat{\mathbf{V}}_{r_k}^\top \mathbf{u}^{(t')}\|_2 &\leqslant (1 - \eta m\theta^2\lambda_{r_k}/2)^{t'}\|\widehat{\mathbf{V}}_{r_k}^\top \mathbf{u}^{(0)}\|_2 + t' \cdot (\eta m\theta^2\lambda_{r_k}/2)^{-1} \cdot (\omega^{1/3}\eta m\theta^2)^2 \cdot \sqrt{n} \cdot \widetilde{\mathcal{O}}(1 + \omega\sqrt{m}) \\
&\quad + (\eta m\theta^2\lambda_{r_k}/2)^{-1} \cdot \widetilde{\mathcal{O}}(\omega^{1/3}\eta m\theta^2) \cdot \|(\widehat{\mathbf{V}}_{r_k}^\perp)^\top \mathbf{u}^{(0)}\|_2 \\
&= (1 - \eta m\theta^2\lambda_{r_k}/2)^{t'}\|\widehat{\mathbf{V}}_{r_k}^\top \mathbf{u}^{(0)}\|_2 + t'\lambda_{r_k}^{-1} \cdot \omega^{2/3}\eta m\theta^2 \cdot \sqrt{n} \cdot \widetilde{\mathcal{O}}(1 + \omega\sqrt{m}) \\
&\quad + \lambda_{r_k}^{-1} \cdot \widetilde{\mathcal{O}}(\omega^{1/3}) \cdot \|(\widehat{\mathbf{V}}_{r_k}^\perp)^\top \mathbf{u}^{(0)}\|_2 \\
&\leqslant (1 - \eta m\theta^2\lambda_{r_k}/2)^{t'}\|\widehat{\mathbf{V}}_{r_k}^\top \mathbf{u}^{(0)}\|_2 + t'\lambda_{r_k}^{-1} \cdot \omega^{2/3}\eta m\theta^2 \cdot \sqrt{n} \cdot \widetilde{\mathcal{O}}(1 + \omega\sqrt{m}) \\
&\quad + \lambda_{r_k}^{-1} \cdot \widetilde{\mathcal{O}}(\omega^{1/3}) \cdot \|(\widehat{\mathbf{V}}_{r_k}^\perp)^\top \mathbf{u}^{(0)}\|_2.
\end{aligned}$$

This completes the proof of (B.2). Finally, for (B.3), by assumption we have $\omega^{1/3}\eta m\theta^2 \leqslant \tilde{\mathcal{O}}(1)$. Therefore

$$
\begin{aligned}
\|\mathbf{u}^{(t'+1)}\|_2 &\leqslant \|[\mathbf{I}-(\eta m\theta^2/n)\mathbf{K}^\infty]\hat{\mathbf{V}}_{r_k}\hat{\mathbf{V}}_{r_k}^\top\mathbf{u}^{(t')}\|_2 + \|[\mathbf{I}-(\eta m\theta^2/n)\mathbf{K}^\infty]\hat{\mathbf{V}}_{r_k}^\perp(\hat{\mathbf{V}}_{r_k}^\perp)^\top\mathbf{u}^{(t')}\|_2 \\
&\quad + \tilde{\mathcal{O}}(\omega^{1/3}\eta m\theta^2)\cdot\|\hat{\mathbf{V}}_{r_k}^\top\mathbf{u}^{(t')}\|_2 + \tilde{\mathcal{O}}(\omega^{1/3}\eta m\theta^2)\cdot\|(\hat{\mathbf{V}}_{r_k}^\perp)^\top\mathbf{u}^{(t')}\|_2 \\
&\leqslant (1-\eta m\theta^2\hat{\lambda}_{r_k})\|\hat{\mathbf{V}}_{r_k}^\top\mathbf{u}^{(t')}\|_2 + \tilde{\mathcal{O}}(\omega^{1/3}\eta m\theta^2)\cdot\|\hat{\mathbf{V}}_{r_k}^\top\mathbf{u}^{(t')}\|_2 + \tilde{\mathcal{O}}(1)\cdot\|(\hat{\mathbf{V}}_{r_k}^\perp)^\top\mathbf{u}^{(t')}\|_2 \\
&\leqslant (1-\eta m\theta^2\lambda_{r_k}/2)\|\hat{\mathbf{V}}_{r_k}^\top\mathbf{u}^{(t')}\|_2 + \tilde{\mathcal{O}}(1)\cdot\|(\hat{\mathbf{V}}_{r_k}^\perp)^\top\mathbf{u}^{(t')}\|_2 \\
&\leqslant (1-\eta m\theta^2\lambda_{r_k}/2)\|\hat{\mathbf{V}}_{r_k}^\top\mathbf{u}^{(t')}\|_2 + \tilde{\mathcal{O}}(1)\cdot\|(\hat{\mathbf{V}}_{r_k}^\perp)^\top\mathbf{u}^{(0)}\|_2 + t'\omega^{1/3}\eta m\theta^2\sqrt{n}\cdot\tilde{\mathcal{O}}(1+\omega\sqrt{m})
\end{aligned}
$$

for $t' = 0,\ldots,t-1$, where the third inequality is by Lemma B.1 and the assumption that $\omega \leqslant \tilde{\mathcal{O}}(\lambda_{r_k}^3)$, and the fourth inequality follows by (B.1). Therefore we have

$$
\|\mathbf{u}^{(t')}\|_2 \leqslant \mathcal{O}(\sqrt{n})\cdot(1-\eta m\theta^2\lambda_{r_k}/2)^{t'} + \tilde{\mathcal{O}}((\eta m\theta^2\lambda_{r_k})^{-1})\cdot\|(\hat{\mathbf{V}}_{r_k}^\perp)^\top\mathbf{u}^{(0)}\|_2 + \lambda_{r_k}^{-1}t'\omega^{1/3}\sqrt{n}\cdot\tilde{\mathcal{O}}(1+\omega\sqrt{m}).
$$

This finishes the proof. $\qquad\square$

# D  AUXILIARY LEMMAS

In this section we list several auxiliary lemmas on the properties of over-parameterized neural networks we need in our proof of Theorem 3.2. These results are mostly summarized from Allen-Zhu et al. (2018) and Cao and Gu (2019a).

## D.1  AUXILIARY LEMMAS

Denote

$$
\begin{aligned}
\mathbf{D}_i &= \operatorname{diag}\big(\mathbb{1}\{(\mathbf{W}_1\mathbf{x}_i)_1 > 0\},\ldots,\mathbb{1}\{(\mathbf{W}_1\mathbf{x}_i)_m > 0\}\big), \\
\mathbf{D}_i^{(0)} &= \operatorname{diag}\big(\mathbb{1}\{(\mathbf{W}_1^{(0)}\mathbf{x}_i)_1 > 0\},\ldots,\mathbb{1}\{(\mathbf{W}_1^{(0)}\mathbf{x}_i)_m > 0\}\big).
\end{aligned}
$$

**Lemma D.1** (Allen-Zhu et al. (2018))**.** If $\omega \leqslant \mathcal{O}([\log(m)]^{-3/2})$, then with probability at least $1-\mathcal{O}(n)\cdot\exp[-\Omega(m\omega^{2/3})]$,

$$
\|\mathbf{D}_i - \mathbf{D}_i^{(0)}\|_0 \leqslant \mathcal{O}(\omega^{2/3}m)
$$

for all $\mathbf{W} \in \mathcal{B}(\mathbf{W}^{(0)},\omega)$, $i \in [n]$.

**Lemma D.2** (Cao and Gu (2019a))**.** There exists an absolute constant $\kappa$ such that, with probability at least $1-\mathcal{O}(n)\cdot\exp[-\Omega(m\omega^{2/3})]$ over the randomness of $\mathbf{W}^{(1)}$, for all $i \in [n]$ and $\mathbf{W},\mathbf{W}' \in \mathcal{B}(\mathbf{W}^{(0)},\omega)$ with $\omega \leqslant \kappa[\log(m)]^{-3/2}$, it holds uniformly that

$$
|f_{\mathbf{W}'}(\mathbf{x}_i) - f_{\mathbf{W}}(\mathbf{x}_i) - \langle\nabla_\mathbf{W}f_\mathbf{W}(\mathbf{x}_i),\mathbf{W}'-\mathbf{W}\rangle| \leqslant \mathcal{O}\Big(\omega^{1/3}\sqrt{m\log(m)}\Big)\cdot\|\mathbf{W}_1'-\mathbf{W}_1\|_2.
$$

**Lemma D.3** (Cao and Gu (2019a))**.** For any $\delta > 0$, if $m \geqslant C\log(n/\delta)$ for a large enough absolute constant $C$, then with probability at least $1-\delta$, $|f_{\mathbf{W}^{(0)}}(\boldsymbol{x}_i)| \leqslant \mathcal{O}(\sqrt{\log(n/\delta)})$ for all $i \in [n]$.

**Lemma D.4** (Cao and Gu (2019a))**.** There exists an absolute constant $C$ such that, with probability at least $1-\mathcal{O}(n)\cdot\exp[-\Omega(m\omega^{2/3})]$, for all $i \in [n]$, $l \in [L]$ and $\mathbf{W} \in \mathcal{B}(\mathbf{W}^{(0)},\omega)$ with $\omega \leqslant C[\log(m)]^{-3}$, it holds uniformly that

$$
\|\nabla_{\mathbf{W}_l}f_\mathbf{W}(\mathbf{x}_i)\|_F \leqslant \mathcal{O}(\sqrt{m}).
$$

The following lemma provides a uniform bound of the neural network function value over $\mathcal{B}(\mathbf{W}^{(0)},\omega)$.

**Lemma D.5.** Suppose that $m \geqslant \Omega(\omega^{-2/3}\log(n/\delta))$ and $\omega \leqslant \mathcal{O}([\log(m)]^{-3})$. Then with probability at least $1-\delta$, $|f_\mathbf{W}(\mathbf{x}_i)| \leqslant \mathcal{O}(\sqrt{\log(n/\delta)} + \omega\sqrt{m})$ for all $\mathbf{W} \in \mathcal{B}(\mathbf{W}^{(0)},\omega)$ $i \in [n]$.

**Lemma D.6.** If $\omega \leqslant \mathcal{O}([\log(m)]^{-3/2})$, then with probability at least $1-\mathcal{O}(n)\cdot\exp[-\Omega(m\omega^{2/3})]$,

$$
\|\nabla_\mathbf{W}f_\mathbf{W}(\mathbf{x}_i) - \nabla_\mathbf{W}f_{\mathbf{W}^{(0)}}(\mathbf{x}_i)\|_F \leqslant \mathcal{O}(\omega^{1/3}\sqrt{m}),
$$

$$
|\langle\nabla_\mathbf{W}f_\mathbf{W}(\mathbf{x}_i),\nabla_\mathbf{W}f_\mathbf{W}(\mathbf{x}_j)\rangle - \langle\nabla_\mathbf{W}f_{\mathbf{W}^{(0)}}(\mathbf{x}_i),\nabla_\mathbf{W}f_{\mathbf{W}^{(0)}}(\mathbf{x}_j)\rangle| \leqslant \mathcal{O}(\omega^{1/3}m)
$$

for all $\mathbf{W} \in \mathcal{B}(\mathbf{W}^{(0)},\omega)$ and $i \in [n]$.

## D.2 Proofs of Lemmas D.5 and D.6

*Proof of Lemma D.5.* By Lemmas D.2 and D.4, we have

$$
\begin{aligned}
|f_{\mathbf{W}}(\mathbf{x}_i) - f_{\mathbf{W}^{(0)}}(\mathbf{x}_i)| &\leqslant \|\nabla_{\mathbf{W}_1} f_{\mathbf{W}^{(0)}}(\mathbf{x}_i)\|_F \|\mathbf{W}_1 - \mathbf{W}_1^{(0)}\|_F + \|\nabla_{\mathbf{W}_2} f_{\mathbf{W}^{(0)}}(\mathbf{x}_i)\|_F \|\mathbf{W}_2 - \mathbf{W}_2^{(0)}\|_F \\
&\quad + \mathcal{O}(\omega^{1/3}\sqrt{m\log(m)}) \cdot \|\mathbf{W}_1 - \mathbf{W}_1^{(0)}\|_2 \\
&\leqslant \mathcal{O}(\omega\sqrt{m}),
\end{aligned}
$$

where the last inequality is by the assumption $\omega \leqslant [\log(m)]^{-3}$. Applying triangle inequality and Lemma D.3 then gives

$$
\begin{aligned}
|f_{\mathbf{W}}(\mathbf{x}_i)| &\leqslant |f_{\mathbf{W}^{(0)}}(\mathbf{x}_i)| + |f_{\mathbf{W}}(\mathbf{x}_i) - f_{\mathbf{W}^{(0)}}(\mathbf{x}_i)| \leqslant \mathcal{O}(\sqrt{\log(n/\delta)}) + \mathcal{O}(\omega\sqrt{m}) \\
&= \mathcal{O}(\sqrt{\log(n/\delta)} + \omega\sqrt{m}),
\end{aligned}
$$

This completes the proof. $\qquad\square$

*Proof of Lemma D.6.* By direct calculation, we have

$$
\nabla_{\mathbf{W}_1} f_{\mathbf{W}^{(0)}}(\mathbf{x}_i) = \sqrt{m} \cdot \mathbf{D}_i^{(0)} \mathbf{W}_2^{(0)\top} \mathbf{x}_i^\top, \nabla_{\mathbf{W}_1} f_{\mathbf{W}}(\mathbf{x}_i) = \sqrt{m} \cdot \mathbf{D}_i \mathbf{W}_2^\top \mathbf{x}_i^\top.
$$

Therefore we have

$$
\begin{aligned}
\|\nabla_{\mathbf{W}_1} f_{\mathbf{W}}(\mathbf{x}_i) - \nabla_{\mathbf{W}_1} f_{\mathbf{W}^{(0)}}(\mathbf{x}_i)\|_F &= \sqrt{m} \cdot \|\mathbf{D}_i \mathbf{W}_2^\top \mathbf{x}_i^\top - \mathbf{D}_i^{(0)} \mathbf{W}_2^{(0)\top} \mathbf{x}_i^\top\|_F \\
&= \sqrt{m} \cdot \|\mathbf{x}_i \mathbf{W}_2 \mathbf{D}_i - \mathbf{x}_i \mathbf{W}_2^{(0)} \mathbf{D}_i^{(0)}\|_F \\
&= \sqrt{m} \cdot \|\mathbf{W}_2 \mathbf{D}_i - \mathbf{W}_2^{(0)} \mathbf{D}_i^{(0)}\|_F \\
&\leqslant \sqrt{m} \cdot \|\mathbf{W}_2^{(0)}(\mathbf{D}_i^{(0)} - \mathbf{D}_i)\|_F + \sqrt{m} \cdot \|(\mathbf{W}_2^{(0)} - \mathbf{W}_2)\mathbf{D}_i\|_F
\end{aligned}
$$

By Lemma 7.4 in Allen-Zhu et al. (2018) and Lemma D.1, with probability at least $1 - n \cdot \exp[-\Omega(m)]$, $\sqrt{m} \cdot \|\mathbf{W}_2^{(0)}(\mathbf{D}_i^{(0)} - \mathbf{D}_i)\|_F \leqslant \mathcal{O}(\omega^{1/3}\sqrt{m})$ for all $i \in [n]$. Moreover, clearly $\|(\mathbf{W}_2^{(0)} - \mathbf{W}_2)\mathbf{D}_i\|_F \leqslant \|\mathbf{W}_2^{(0)} - \mathbf{W}_2\|_F \leqslant \omega$. Therefore

$$
\|\nabla_{\mathbf{W}_1} f_{\mathbf{W}}(\mathbf{x}_i) - \nabla_{\mathbf{W}_1} f_{\mathbf{W}^{(0)}}(\mathbf{x}_i)\|_F \leqslant \mathcal{O}(\omega^{1/3}\sqrt{m})
$$

for all $i \in [n]$. This proves the bound for the first layer gradients. For the second layer gradients, we have

$$
\nabla_{\mathbf{W}_2} f_{\mathbf{W}^{(0)}}(\mathbf{x}_i) = \sqrt{m} \cdot [\sigma(\mathbf{W}_1^{(0)} \mathbf{x}_i)]^\top, \nabla_{\mathbf{W}_2} f_{\mathbf{W}}(\mathbf{x}_i) = \sqrt{m} \cdot [\sigma(\mathbf{W}_1 \mathbf{x}_i)]^\top
$$

It therefore follows by the 1-Lipschitz continuity of $\sigma(\cdot)$ that

$$
\|\nabla_{\mathbf{W}_2} f_{\mathbf{W}}(\mathbf{x}_i) - \nabla_{\mathbf{W}_2} f_{\mathbf{W}^{(0)}}(\mathbf{x}_i)\|_F \leqslant \sqrt{m} \cdot \|\mathbf{W}_1 \mathbf{x}_i - \mathbf{W}_1^{(0)} \mathbf{x}_i\|_F \leqslant \omega\sqrt{m} \leqslant \omega^{1/3}\sqrt{m}.
$$

This completes the proof of the first inequality.

The second inequality directly follows by triangle inequality and Lemma D.4:

$$
\begin{aligned}
|\langle \nabla_{\mathbf{W}} f_{\mathbf{W}}(\mathbf{x}_i), \nabla_{\mathbf{W}} f_{\mathbf{W}}(\mathbf{x}_j)\rangle - m\mathbf{K}^{(0)}| &\leqslant |\langle \nabla_{\mathbf{W}} f_{\mathbf{W}}(\mathbf{x}_i) - \nabla_{\mathbf{W}} f_{\mathbf{W}^{(0)}}(\mathbf{x}_i), \nabla_{\mathbf{W}} f_{\mathbf{W}}(\mathbf{x}_j)\rangle| \\
&\quad + |\langle \nabla_{\mathbf{W}} f_{\mathbf{W}^{(0)}}(\mathbf{x}_i), \nabla_{\mathbf{W}} f_{\mathbf{W}}(\mathbf{x}_j) - \nabla_{\mathbf{W}} f_{\mathbf{W}^{(0)}}(\mathbf{x}_j)\rangle| \\
&\leqslant \mathcal{O}(\omega^{1/3}m).
\end{aligned}
$$

This finishes the proof. $\qquad\square$

# E Additional Experimental Results

## E.1 Estimating the Projection Length in Function Space

As mentioned in Section 4.1, when using freshly sampled points, we are actually estimating the projection length of residual function $r(\mathbf{x}) = f^*(\mathbf{x}) - \theta f_{\mathbf{W}^{(t)}}(\mathbf{x})$ onto the given Gegenbauer polynomial $P_k(\mathbf{x})$. Here we present in Figure 3 a comparison between the projection length using training data and that using test data. An interesting phenomenon is that the network generalizes well

on the lower-order Gegenbauer polynomial and along the highest-order Gegenbauer polynomial the network suffers overfitting.

(a) projection onto training data

(b) projection onto test data

(c) projection onto training data

(d) projection onto test data

Figure 3: Convergence curve for projection length onto vectors (determined by training data) and functions (estimated by test data). We can see that for low-order Gegenbauer polynomials, the network learns the function while for the high-order Gegenbauer polynomial, the network overfits the training data.

## E.2 NEAR-LINEAR CONVERGENCE BEHAVIOR

In this subsection, we present the same curve shown in Section 4.1 in logarithmic scale instead of linear scale. As shown in Figure 4 we can see that the convergence of different projection length is close to linear convergence, which is well-aligned with our theorem. Note that we performed a moving average of range 20 on these curves to avoid the heavy oscillation especially in late stage.

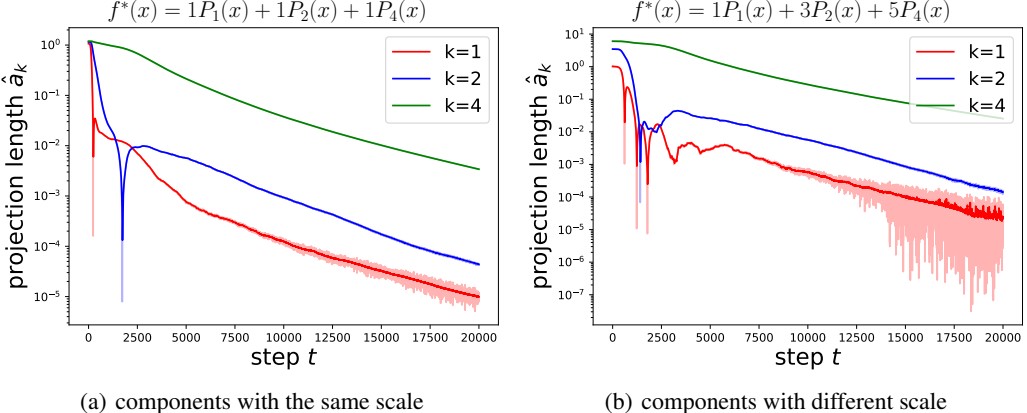

(a) components with the same scale  components with different scale

Figure 4: Log-scale convergence curve for projection length onto different component. (a) shows the curve when the target function have different component with the same scale. (b) shows the curve when the higner-order components have larger scale. Both exhibit nearly linear convergence especially at late stage.

