# OpenReview forum: "Towards Understanding the Spectral Bias of Deep Learning"
_ICLR.cc/2020/Conference — Reject_

### Official Review · AnonReviewer2 · 2019-10-15
**Official Blind Review #2**

**Rating:** 3

**Review:**

This paper studies the training of overparametrized neural networks by gradient descent. More precisely, the authors consider the neural tangent regime (NTK regime). That is, the weights are chosen sufficiently large and the neural network is sufficiently overparametrized. It has been observed that in this scenario, the neural network behaves approximately like a linear function of its weights.

In this regime, the authors show that, the directions corresponding to larger eigenvalues of the neural tangent kernel are learned first. As this corresponds to learning lower-degree polynomials first, the authors claim that this explains the "spectral bias" observed in previous papers.

-I think that from a mathematical point of view, the main result of this paper is what one would expect intuitively:
When performing gradient descent with quadratic loss where the function to be learnt is linear, it is common knowledge that convergence is faster on directions corresponding to larger singular values. Since in the NTK regime, the neural network can be approximated by a linear function around the initialization one expects the behavior predicted by the main results. From a theoretical perspective, I see the main contribution of the paper as making this statement precise.

-I am skeptical about some of the implications for practitioners, which are given by the authors:
For example, on p.5 the authors write "Therefore, Theorem 3.2 theoretically explains the empirical observations given in Rahaman et al. (2018), and demonstrates that the difficulty of a function to be learned by neural network should be studied in the eigenspace of neural tangent kernel." To the best of my knowledge, it is unclear whether practitioners train neural networks in the NTK regime (see, e.g., [1]). Moreover, I am wondering whether some of the assumptions of their theorem are really met in practice. For example, the required sample size for higher order polynomials grows exponentially fast with the order and the required step size goes to zero exponentially fast. Does this really correspond to what is observed in practice? (Or is this a mere artifact of training in the NTK regime?)  Is this what one observes in the experiments by Ramahan?

I think the paper is not yet ready for being published.
 1. There are many typos. Here is an (very incomplete) list.
    -p. 2: "Su and Yang (2019)" improves the convergence..."
    -p. 2: "This theorem gives finer-grained control on error term's"
    -p. 2: "We present a more general results"
    -p. 4: "The variance follows the principal..."
    -p. 4: "...we will present Mercer decomposition in (the) next section."
2. I think that the presentation can be polished and many statements are somewhat unclear. For example, on p. 7 the authors write "the convergence rates [...] are exactly predicted by our theory in a qualitative sense."
    The meaning of this sentence is unclear to me. Does that mean in a quantitative sense? To be honest, only considering Fig. 1 I am not able to assess whether the convergence rates of the different components are truly linear.

I decided for my rating of the paper because of the following reasons:
-I think that for a theory paper the results obtained by the authors are not enough, as they are rather direct consequences of the "near-linearity" of the neural network around the initialization.
-In my view, there is a huge gap between current theoretical results for deep learning and practice. For this reason, it is not problematic for me that it is unclear, what the results in this paper mean for practitioners. (Apart from that, results for the NTK regime are interesting in its own right.) However, in my view, one should explain the limitations of the theory more carefully.
-The presentation of the paper needs to be improved.

References:
[1] A note on lazy training in supervised differentiable programming. L Chizat, F Bach - arXiv preprint arXiv:1812.07956, 2018



-----------------------------

I highly appreciate the authors' detailed response. However, I feel that the paper does not contain enough novelty to justify acceptance.

------
"Equation (8) in Arora et al., (2019b) only provides a bound on the whole residual vector, i.e., , and therefore cannot show different convergence rates along different directions."

When going through Section 4 , I think that it is implicitly stated that one has different convergence along different directions.
-----
For this reason, I am not going to change my score.


**Experience Assessment:**

I do not know much about this area.

**Review Assessment: Checking Correctness Of Derivations And Theory:**

I did not assess the derivations or theory.

**Review Assessment: Checking Correctness Of Experiments:**

I assessed the sensibility of the experiments.

**Review Assessment: Thoroughness In Paper Reading:**

I read the paper at least twice and used my best judgement in assessing the paper.

---

> ### Author Response · Authors · 2019-11-08
> **Response to Reviewer #2**
>
> Thank you for you detailed and helpful comments. We address your questions as follows.
>
> Q1. “from a mathematical point of view… making this statement precise.”
> A1. We agree that our result is intuitive. But we believe this is an advantage of our result, instead of a weak point. We also would like to point out that although the results match intuition, the proof is by no means trivial, especially because it only relies on milder over-parameterization conditions to learn the components of the target function with lower complexity. Given the fact that such a result has not been shown in previous work, closing this gap between mathematical intuition and rigorous theoretical analysis is indeed one of our contributions.
>
>
> Q2. “I am skeptical about some of the implications for practitioners… the NTK regime”
> A2. Thanks for pointing it out. Our analysis is indeed in the NTK regime. However, we would like to emphasize that by focusing on the low complexity components of the target function, we have greatly improved the over-parameterization condition in standard results in the NTK regime (Du et al. (2018b)). Therefore, we believe our work helps pushing the study of neural networks in the NTK regime towards a more practical setting. We would also like to point out that the optimization method studied in this paper is standard gradient descent with a practically used initialization method. For these reasons, we believe that our results are also of great practical value. In the revision, we have emphasized that our analysis is in the NTK regime. However, we believe that the spectral bias phenomenon can be rigorously proved in other regimes of neural network training.
>
>
> Q3.(a). “whether some of the assumptions of their theorem are really met in practice. For example, the required sample size for higher order polynomials grows exponentially fast with the order and the required step size goes to zero exponentially fast.”
> A3.(a). Thanks for your question. We have added a remark (Remark 3.9) to explain such exponential dependency. Here we would like to emphasize that instead of checking the rate in terms of $k$, a more reasonable measure should probably be the relation between $n$ and the number of independent function components being learned, which is $r_k$. Intuitively speaking, based on $n$ samples, it is only reasonable to expect learning less than or equal to $n$ independent components of the true function, and the exponential dependency in $k$ is a natural consequence of the fact that in the high dimensional space, there are a large number of linearly independent polynomials even for very low degrees. From this we can see that $n \geq r_k = \Omega(d^{k-1})$ is not an artifact, and is a reasonable and unavoidable assumption. In fact, even if we know that the target function is exactly a polynomial with degree less than or equal to $k$, it still requires exponentially many samples to fit this polynomial, since the number of coefficients in a high-dimensional polynomial is exponential in the degree of the polynomial.
>
>
> Q3.(b). “Does this really correspond to what is observed in practice? (Or is this a mere artifact of training in the NTK regime?)  Is this what one observes in the experiments by Ramahan?”
> A3.(b). The effect of different sample sizes are not considered in the experiments in Rahaman et al. (2018), and therefore no empirical observation conflicts with our theory. In fact, the discussion in Rahaman et al. (2018) below Theorem 1 actually matches our calculation in the setting $k \gg d$, which backs up our theory. Moreover, we would also like to emphasize that all our results regarding spherical harmonics and the exponential dependency in their degrees are only a special case of Theorem 3.2 when the data inputs are uniformly sampled from unit sphere. Such exponential dependency does not necessarily exist for other input distributions.
>
>
> Q4. About typos and presentation
> A4. Thank you for pointing out these typos and presentation issues. We apologize for the typos and unclear statements. We have improved the presentation of our paper and fixed typos in the revision.
>
>
> Q5. “only considering Fig. 1… convergence rates of the different components are truly linear.”
> A5. We admit that the figure under linear scale is not enough to show the linear convergence rate, and have added the same curves in log scale in Appendix E.2. In log scale we can now see that the curves indeed demonstrate linear convergence.
>
>
> Q6. “in my view, one should explain the limitations of the theory more carefully.”
> A6. Thanks for your suggestion. We have rephrased Remark 3.3 and mentioned in Section 1 and Section 5 to make it clear that there is still a gap between theory and practice in terms of the spectral bias of neural networks.
>
>
> We hope you find your concerns satisfactorily addressed by our response, which has also been reflected in the revised paper. Please let us know if you have more comments or any other suggestions.

---

### Official Review · AnonReviewer1 · 2019-10-16
**Official Blind Review #1**

**Rating:** 6

**Review:**

I must qualify my review by stating that I am not an expert in kernel methods, and the mathematics in the proof is more advanced than I typically use. So it is possible that there are technical flaws to this work that I did not notice.

That being said, I found this to be quite an interesting paper. It provides a concise explanation for the types of features learned by ANNs: those that correspond to the largest eigenvalues of the kernel function. Because these typically correspond to the lowest-frequency components, this means that the ANNs tend to first learn the low frequency components of their target functions. This provides a nice explanation for how ANNs can both: a) have enough capacity to memorize random data; yet b) generalize fairly well in many tasks with structured input data. In the case of structured data, there are low frequency components that correspond to successfully generalized solutions.

I have a few questions about the generality of this result, and its application to make better machine learning systems:

1) As far as I can tell, the proof applies strictly vanilla SGD (algorithm 1). Would it be possible to extend this proof to other optimizers (say, ADAM)? That extension would help to connect this theory to the practical side of the field.

2)  Given that the kernel depends on the loss function, and it's the eigenspectrum of the kernel's integrator operator that determines the convergence properties, can this work be applied to engineering better loss functions for practical applications?


**Experience Assessment:**

I do not know much about this area.

**Review Assessment: Checking Correctness Of Derivations And Theory:**

I did not assess the derivations or theory.

**Review Assessment: Checking Correctness Of Experiments:**

I assessed the sensibility of the experiments.

**Review Assessment: Thoroughness In Paper Reading:**

I made a quick assessment of this paper.

---

> ### Author Response · Authors · 2019-11-08
> **Response to Reviewer #1**
>
> Thank you for your insightful comments! We address your questions as follows:
>
> Q1. “As far as I can tell, the proof applies strictly vanilla SGD... practical side of the field.”
> A1. Our current analysis focuses on vanilla gradient descent. However, we believe that other practically useful algorithms like ADAM should exhibit similar spectral bias phenomenon.  Combining our current analysis with the analysis in Wu et al. (2019) and Zhou et al. (2018) can potentially provide similar result for ADAM-type algorithms, and this can be an interesting and promising future work direction. We have added some discussion in Section 5.
>
>
> Q2. “Given that the kernel depends on the loss function, and it's the eigenspectrum of the kernel's integrator operator that determines the convergence properties, can this work be applied to engineering better loss functions for practical applications?”
> A2. Thank you for your great question. We would like to clarify that the definition of the neural tangent kernel should be independent of the loss functions. On the other hand, the kernel function actually depends on the neural network architecture. This suggests that our work might be applied to engineering better architectures, or compare different architectures. For example, comparison on spectral properties of the kernels corresponding to ResNets, CNNs and fully connected networks may shed light on the design of more effective network architectures.
>
>
> We hope that our answers have addressed your questions. We also revised our paper accordingly. Any further comments on the paper are more than welcome.

---

### Official Review · AnonReviewer3 · 2019-10-21
**Official Blind Review #3**

**Rating:** 6

**Review:**

The paper aims to provide theoretical justification for a "spectral bias" that is observed in training of neural networks: a phenomenon recorded in literature (Rahaman et al.), where lower frequency components of a signal are fit faster than higher frequency ones. The contributions of the paper are as follows:
1. Proves an upper bound on the rate of convergence on the residual error projected on top few eigenfunctions (of a certain integral operator). The upper bound is in terms of the eigenvalues of the corresponding eigenfunctions and is distribution independent.
2. Provides an upper bound on the decay of eigenvalues in the case of depth-2 ReLU networks and also a exact characterization of the eigenfunctions. While such upper bounds and the characterization of eigenfunctions existed in literature earlier, it is argued that the new bounds are better.
3. Combining the above two results, a justification is obtained for the "spectral bias" phenomenon that is recorded in literature.
4. Some toy experiments are provided to exhibit the spectral bias phenomenon.

Recommendation:
I recommend "weak acceptance". The paper takes a step towards explaining the phenomenon of spectral bias in deep learning. While concrete progress is made in the context of depth-2 ReLU networks (even though in NTK regime), perhaps the ideas could be extended to deeper networks.

Technical comments:
- It is argued that the new bound of $O(\mathrm{min}(k^{-d-1}, d^{-k+1}))$ is better than the bound of $O(k^{-d-1})$ from the previous work of Bietti and Mairal, in the regime where $d \gg k$. I think there is a typo here. In the regime of $d \gg k$, the bound $k^{-d-1}$ is the smaller one so both bounds are comparable. It is argued that $d \gg k$ is the more relevant regime, but then there isn't any improvement here.
- The proof of spectral analysis is said to follow a similar outline as compared to the prior work of Bietti-Mairal, but it is not clear to me where this new proof deviates and improves on prior techniques? Or is it just a more careful analysis of the prior techniques?
- The proof operates in the "Neural Tangent Kernel" regime, by considering hugely overparameterized networks. This can be viewed as a negative thing, but then, most results in literature also operate in this regime and it is a major challenge for the field to prove results in the mildly overparameterized / non-NTK regime!

Potential suggestions for improvement:
- In Section 4: the y-axis of the graph is labeled "error's coefficient" which is non-informative. Is it $|a_k - \hat{a}_k|$ ? I also had a question here about the proposed Nystrom method: Why is it okay to use the training points in the Nystrom method. Ideally, we should use freshly sampled points. Is there a justification for using the training points? If not, perhaps it is best to go with freshly sampled points.
- I felt the proofs in the Appendix are very opaque and it is hard to pinpoint what the new insight is (at least for a reader, like me, who does not have an in-depth familiarity with these convergence proofs).


**Experience Assessment:**

I have read many papers in this area.

**Review Assessment: Checking Correctness Of Derivations And Theory:**

I assessed the sensibility of the derivations and theory.

**Review Assessment: Checking Correctness Of Experiments:**

I assessed the sensibility of the experiments.

**Review Assessment: Thoroughness In Paper Reading:**

I read the paper at least twice and used my best judgement in assessing the paper.

---

> ### Author Response · Authors · 2019-11-08
> **Response to Reviewer #3**
>
> Thank you very much for your helpful and positive comments. We address your questions as follows.
>
> Q1. “It is argued that the new bound of... there isn't any improvement here.”
> A1. Thank you very much for pointing out this issue. We believe that this is a misunderstanding caused by a typo and a misuse of the big-O notation, which we have fixed in the revision. $\mu_k$ should be $\Omega(\max(k^{-d-1}, d^{-k+1}))$ instead of the minimum of the two terms. Since the convergence speed is characterized as $(1-\mu_k)^t$, the larger $\mu_k$ is, the faster gradient descent converges. We can see that our result is better then previous results when $d \gg k$, since we provide a larger lower bound for $\mu_k$.
>
>
> Q2. “The proof of spectral analysis is said to follow a similar outline... prior techniques?”
> A2. Our proof uses the same technique as the proof of Bietti and Mairal (2019) for the $k \gg d$ setting. The major difference between our result and Bietti and Mairal (2019)’s is that we also consider the case $d \gg k$, which is a more practical setting. This leads to an improved characterization for the eigenvalue $\mu_k$. We have emphasized the difference in Remark 3.6.
>
>
> Q3. “The proof operates in... the mildly overparameterized / non-NTK regime!”
> A3. By far, we only focus on the NTK regime and extend previous results by presenting a more precise characterization of convergence result. However we would also like to emphasize that in Theorem 3.2, the over-parameterization requirement is only related to $\lambda_{r_k}$, the $k$-th distinct eigenvalue of NTK. Therefore our theory indeed works for networks with milder over-parameterization, compared with many prior results (for example, Du et al. (2018b)). We have emphasized this in Remark 3.3. Analysis in non-NTK regime is beyond the scope of this paper, and can be an interesting future work direction.
>
>
> Q4. “In Section 4: the y-axis of the graph... freshly sampled points.”
> A4. Thanks for the suggestion. We have changed the ‘error’s coefficient’ into ‘projection length’ and given a clearer definition. Ideally the error coefficient is the Gegenbauer coefficient of the residual function: $f^*(x) - \theta f_{\mathbf{W}^{(t)}}(x)$, which can be seen as the projection length onto the Gegenbauer polynomial.
>
> The experiments are designed to demonstrate the result of our main theorem, which states that the residual of training data, projected to certain eigenfunctions, will decrease at a certain linear rate depending on the eigenvalues. So what we want to show by the experiments is merely about how the residual of training data behaves. That is actually the projection length onto the vectors defined by Gegenbauer polynomials. We admit that the original y-axis label and the word ‘Nystrom’ is not very appropriate. We have presented a more informative definition in our revised version.
>
> In the case where freshly sampled points are used, what we can get following the same procedure is the residual function’s Gegenbauer coefficient, which can be seen as the projection length in function space. We also present these results in Appendix E.1
>
>
> Q5. “I felt the proofs in the Appendix are very opaque... these convergence proofs).”
> A5. We apologize for the unclear proofs. To improve readability, we have added more comments before each lemma in Section B.1 about the intuition and the role of these lemmas in the main proof.
>
>
> The response above has been reflected in our revised paper. Please let us know if you have any further suggestions.

---

### Decision · Program_Chairs · 2019-12-19

**Decision:**

Reject

**Comment:**

The authors propose to understand spectral bias during training of neural networks from the perspective of the NTK. While reviewers appreciated aspects of the work, the general consensus was that the current version is not ready for publication; some concerns stem from whether the the NTK model and finite neural networks are sufficiently similar that we should be able to gain real practical insights into the behaviour of finite models. This is partly an empirical question, and stronger experiments are required to have a better sense of the answer. Nonetheless, the authors are encouraged to persist with this work, taking into account reviewer comments in future revisions.